# Migratory lifestyle carries no added overall energy cost in a partial migratory songbird

Nils Linek [1,2,5] ✉, Scott W. Yanco [3,4,5], Tamara Volkmer [1,2], Daniel Zuñiga[1,2], Martin Wikelski [1,2] & Jesko Partecke [1,2]

Seasonal bird migration may provide energy benefits associated with moving to areas with less physiologically challenging climates or increased food availability, but migratory movements themselves may carry high costs. However, time-dynamic energy profiles of free-living migrants—especially small-bodied songbirds—are challenging to measure. Here we quantify energy output and thermoregulatory costs in partially migratory common blackbirds using implanted heart rate and temperature loggers paired with automated radio telemetry and energetic modelling. Our results show that blackbirds save considerable energy in preparation for migration by decreasing heart rate and body temperature 28 days before departure, potentially dwarfing the energy costs of migratory flights. Yet, in warmer wintering areas, migrants do not appear to decrease total daily energy expenditure despite a substantially reduced cost of thermoregulation. These findings indicate differential metabolic programmes across different wintering strategies despite equivalent overall energy expenditure, suggesting that the maintenance of migration is associated with differences in energy allocation rather than with total energy expenditure.

Seasonal bird migration is an impressive and widespread phenomenon[1] that evolves primarily to capitalize on ubiquitous environmental seasonality[2,3]. In temperate environments, the onset of winter brings a decrease in available energy supplies[4], along with an increase in the energy cost of thermoregulation. Thus, the net energy expenditure required to ensure winter survival increases relative to other seasons[5,6] and, for some species, favours escape to milder regions through migration. Although active travel during migration can be energetically costly[7], theory predicts that migration confers other benefits, such as milder weather conditions, greater food availability or reduced predation[1,8,9]. Many of these benefits may directly offset the metabolic demands of migration itself, while others might necessitate alternative life history strategies to overcome energy deficits. However, the specifics of if, when and how migrants realize the presumed energy benefits of their mobile lifestyle remain unknown because it was previously impossible to quantify the dynamic energy consumption of free-living migratory individuals over multiple seasons.

The common blackbird (*Turdus merula*) is a wide-ranging species across Europe and has populations with varying proportions of migratory individuals[10]. Blackbirds from our study population share a common breeding area in southern Germany, from which roughly 25% of birds migrate to winter in southern Europe each year ('migrants'), 66% remain on the breeding grounds year-round ('residents') and a further 9% leave mid-winter when declining temperatures lead to ground frost and continuous snowfall covers the ground, which starkly decreases food availability ('winter escapees')[10,11] (Fig. 1a). In recent years warming temperatures[12] and increasing urbanization[13] have led to a decrease in migratory propensity in the species which opens various questions about drivers and energetic consequences of migration. Migrants winter on average 793 km (median, minimum 275 km, maximum 1,717 km) south-west of the breeding site[14] (Fig. 1b) and over 39 consecutive years have experienced on average -5.7 °C warmer ambient temperatures ($T_a$) ($t = −21.56$, d.f. 60.65, $P < 0.01$) than their resident counterparts throughout the non-breeding season (Fig. 1c).

[1]Max Planck Institute of Animal Behavior, Radolfzell, Germany. [2]Department of Biology, University of Konstanz, Konstanz, Germany. [3]Center for Biodiversity and Global Change, Yale University, New Haven, CT, USA. [4]Department of Ecology and Evolutionary Biology, Yale University, New Haven, CT, USA. [5]These authors contributed equally: Nils Linek, Scott W. Yanco. ✉e-mail: nlinek@ab.mpg.de

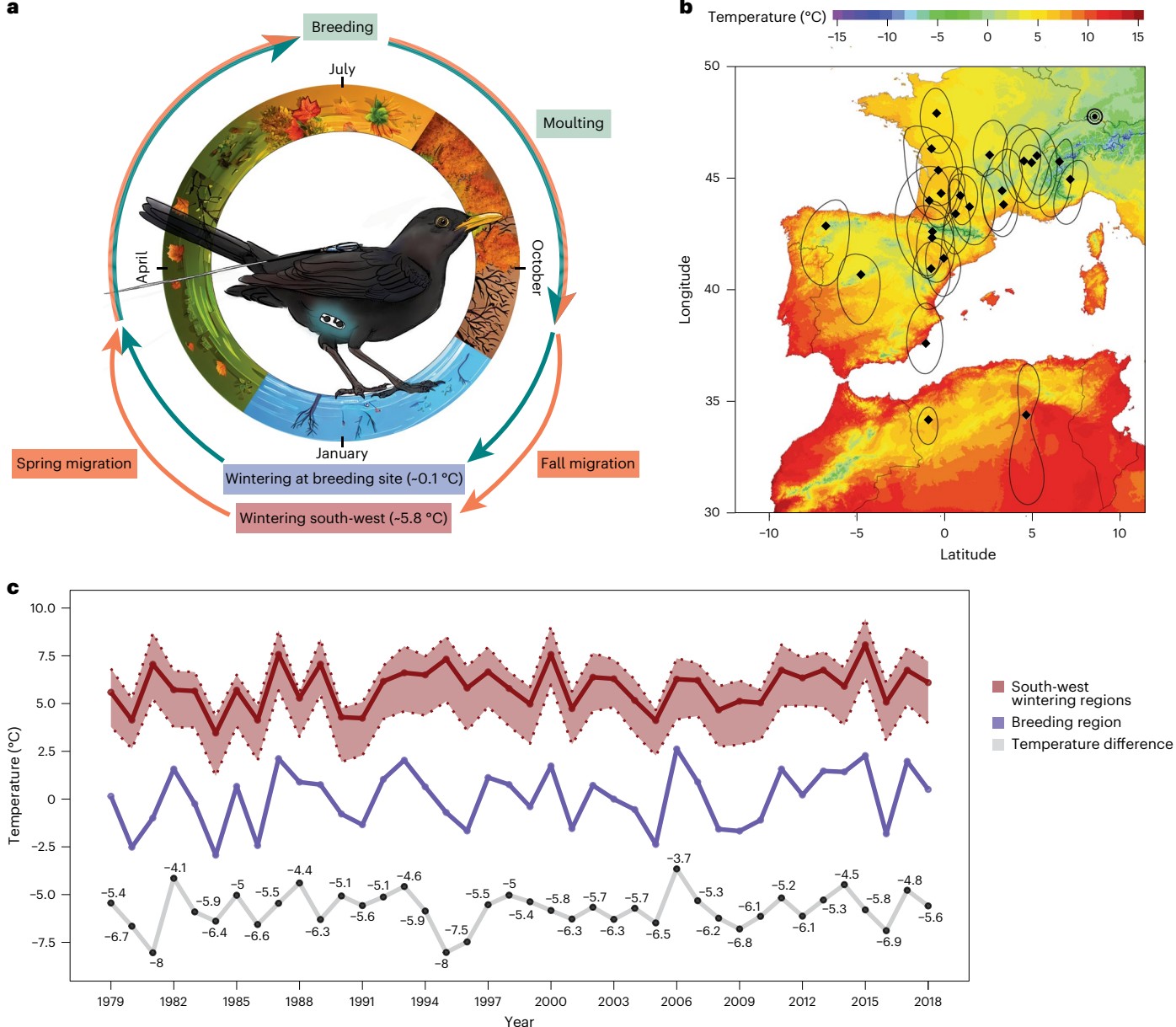

**Fig. 1 | Study system and temperature conditions. a**, Illustration of the experimental setup with a common blackbird (*Turdus merula*) carrying a radio transmitter backpack and an implanted $f_H$ and temperature logger. The surrounding seasonal cycle highlights the main phases during the year for both wintering strategies. **b**, Temperature map for south-west Europe with known breeding and wintering sites of previously studied migratory blackbirds ($N = 25$) of the same population as the birds in the current study. The temperature gradient represents the mean $T_a$ during December and January in south-west Europe. The black triple circle depicts the breeding site and single black diamonds and black outlines (25% kernel utilization distribution) represent the centroid of wintering sites estimated by using geolocators of blackbirds from the same breeding area from a previous study[14]. **c**, Comparison of temperatures between wintering sites and breeding site during winter. The mean $T_a$ during winter (3 December to 17 January) at wintering sites (red, including the lower 25th and upper 75th quantiles) and at the breeding site (blue) over 39 years. The grey line underneath represents the mean temperature difference and calculated value between both location types.

In this study, we aim to assess phenotype-specific differences in metabolic programmes. First, we examined whether overall heart rate ($f_H$), a proxy for energy expenditure[15,16], differs between migrant and resident blackbirds. Second, we investigated whether energy or thermoregulatory dynamics differ among phenotypes (that is, migrants versus residents). Lastly, we quantified if those differences imply differential energy allocation among organismal processes. Previous work in blackbirds has shown that $f_H$ negatively correlated with $T_a$ (ref. 17). Therefore, we presumed warmer $T_a$ on wintering sites to reduce total energy expenditure (that is, $f_H$) of migrant birds via reduced metabolic costs of thermoregulation[18]. We, thus, predicted that migrants would

on average exhibit ~7% lower $f_H$ than residents during winter based on a previously estimated relationship between $f_H$ and $T_a$ in resident individuals[17]. Conversely, we expected migrants to bear increased energy costs of previously unquantified magnitude due to migration itself.

To measure the relative energy costs and benefits of migratory versus sedentary lifestyles, we compared individual blackbirds' $f_H$ and core body temperature ($T_b$) throughout fall, winter and spring. We measured $f_H$ and $T_b$ at 30 min intervals spanning the entire non-breeding season (starting before fall migration and ending after spring migration) for individual resident ($N = 54$) and migrant ($N = 19$) blackbirds using surgically implanted miniature bio-loggers (Star-Oddi, DST micro-HRT,

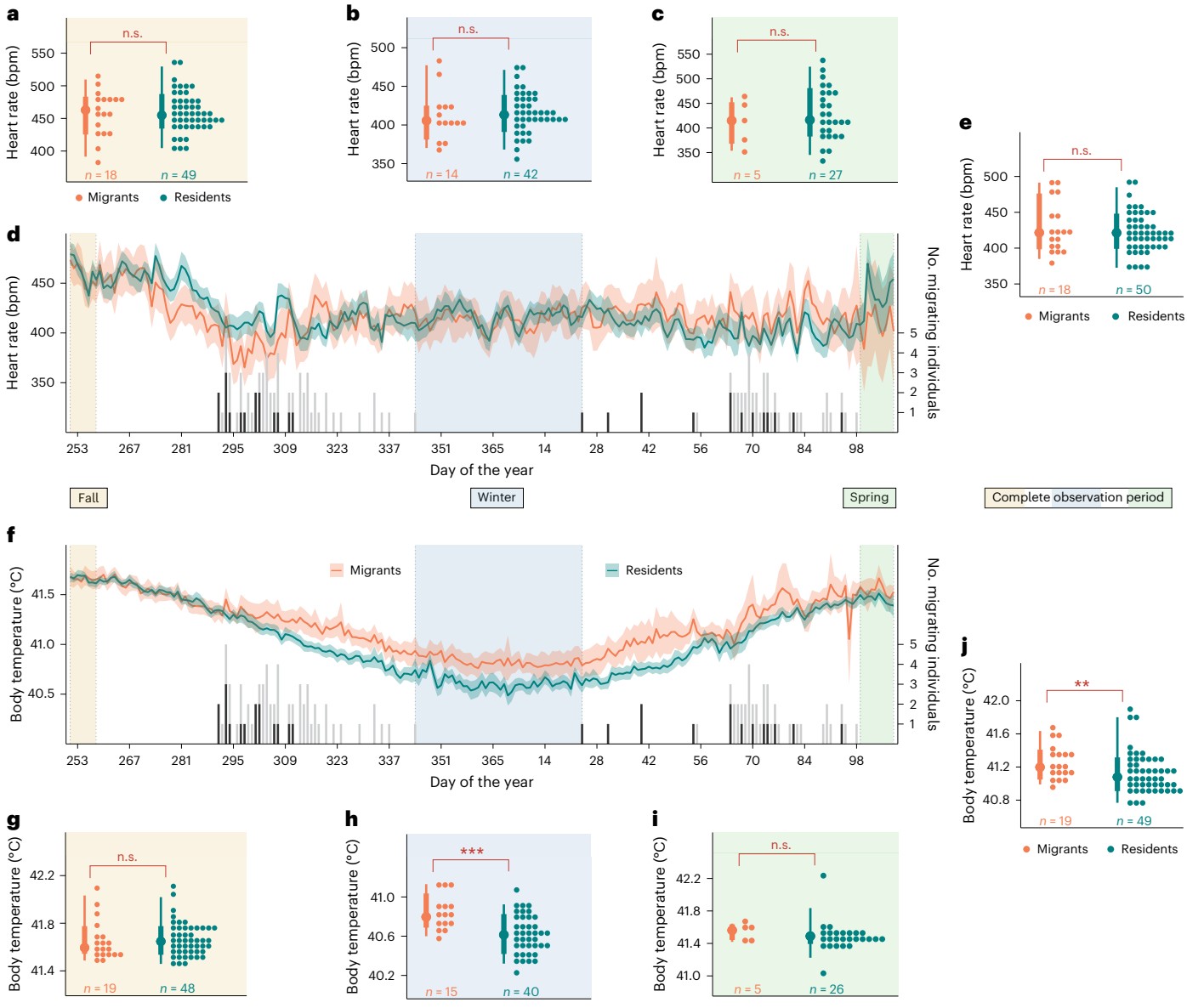

**Fig. 2 | Temporal comparison of $f_H$ and $T_b$ between overwintering strategies with depicted individual migration events. a–j,** The mean $f_H$ (**a–e**) and $T_b$ (**f–j**) over time (**d** and **f**) and during distinct time periods (**a**, **b**, **c**, **e** and **g–j**) are displayed for both wintering strategies with 95% confidence intervals. The black/grey histograms mark the number of individuals migrating each night: the black bars depict the number of individuals on their first night of migration and the grey bars show the number of individuals on subsequent migration nights (right $y$ axis). The ochre time frame in the first week of the experiment highlights the fall period that precedes the initial departures by at least 30 days. The middle blue area between the last fall migration event and the first spring migration marks the core winter period, while the green marked period defines the spring period, which starts with the return of the last migrant to the breeding area. The dots mark individual means in fall for **a** and **g**, winter for **b** and **h**, spring for **c** and **i** and the whole timeframe for **e** and **j**, next to the coloured bars showing distribution within each wintering strategy (mean, and 75% and 25% percentiles). Sample sizes are shown below each group. Significant differences, derived from a linear mixed model with Bonferroni correction (Supplementary Table 1 and Supplementary Results) are indicated by asterisks: ***$P < 0.001$, **$P < 0.01$, *$P < 0.05$ and 'non-significant (n.s.)' where $P > 0.05$.

8.3 × 25.4 mm, 3.3 g; Fig. 1a). Finally, we quantified differences in $f_H$ and $T_b$ and modelled expected thermoregulatory energy expenditure among wintering strategies across seasons (Fig. 2), as well as during eight individual-specific periods representing key migratory stages (Fig. 3).

## Equality of overall energetics

Overall, $f_H$ did not differ between residents and migrants (estimate (EST) of 0.94, standard error (SE) of 6.82, $Z = 0.14$, $P = 0.891$; Fig. 2e and Supplementary Table 1), even during winter (EST of −9.66, SE of 7.89, $Z = −1.22$, $P = 0.221$; Fig. 2b and Supplementary Table 1), when differences in $T_a$ are assumed to be most pronounced (Fig. 1c). Thus,

wintering in warmer locations apparently did not change migrants' overall energy expenditure relative to residents. Moreover, migratory blackbirds exhibited a slightly (but significantly) higher $T_b$ than residents (EST of 0.11, SE of 0.04, $Z = 2.92$, $P = 0.003$; Fig. 2j and Supplementary Table 1), particularly while occupying warmer wintering sites (EST of 0.18, SE of 0.04, $Z = 3.98$, $P < 0.001$; Fig. 2h and Supplementary Table 1). The magnitude of the difference in $T_b$ between migrants and residents was almost the same as the -0.14 °C warmer $T_b$ we expected based on an earlier study[17].

Given that the difference between $T_b$ and $T_a$ was larger for residents than for migrants and considering that heat loss intensifies with greater

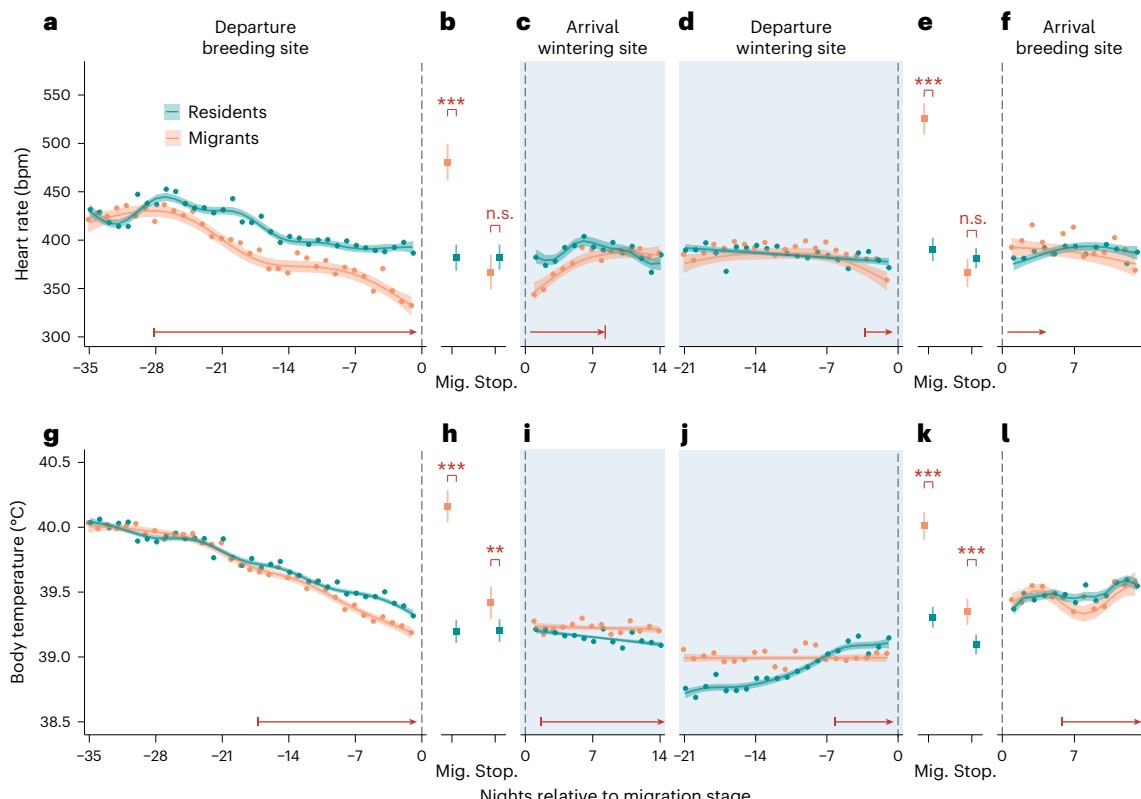

**Fig. 3 | $f_H$ and $T_b$ in different stages relative to migration. a–f,** Mean $f_H$ (**a–e**) and $T_b$ (**g–l**) in seperate chronological stages in relation to migration, are shown as points during night across all migrants (orange) and all residents (dark green) centred on departure date from breeding site relative to initial departure (**a** and **g**), migration (Mig.) and stopover (Stop.) (**b** and **h**) in fall, centred on arrival date in wintering site (**c** and **i**), centred on departure date from wintering site relative to spring departure (**d** and **j**), migration and stopover in spring (**e** and **k**) and centred on arrival date on breeding site (**f** and **l**). For all measurements over time (**a, c, d, f, g** and **i–l**), the vertical dashed line marks the point of reference, while each single point represents the mean value across each overwintering strategy, with migrants centred and residents correspondingly assigned (Methods). The coloured solid line shows predicted $f_H$ and $T_b$ values for each strategy derived from a GAMM, including individual measurements for each bird (Supplementary Tables 2–7 and Supplementary Results). Correspondingly coloured ribbons show the 95% confidence interval of those predictions. The blue-marked periods highlight the time when migratory birds reside in their final wintering grounds. The horizontal red arrows mark the first and last times when measures significantly differ between strategies. For migration stage-centred comparisons via linear mixed models (**b, e, h** and **k**), the means are shown as coloured squares with standard error bars. Bonferroni corrected statistical significance levels: \*\*\*$P < 0.001$, \*\*$P < 0.01$, \*$P < 0.05$ and 'non-significant (n.s.)' where $P > 0.05$.

$T_b − T_a$ differences[19], the actual energy expenditure on thermoregulation probably varies between the two strategies. On the other hand (and unsurprisingly), migratory movements themselves incurred energetic costs (as expressed by $f_H$) for migratory individuals that resident birds did not experience. Together, these findings imply that while overall energy expenditure apparently does not vary between the two strategies, allocation of energy to specific organismal processes probably vary between migrants and residents during various phases of migration and overwintering (for example, migration preparation, stopover and arrival).

## Metabolic dynamics and cost of migration

Migratory travel itself can be energetically expensive[7,20] and often requires special adjustments in physiological processes with changes in the physical makeup, for example, size and weight, of organs and tissue[21,22]. Migrants' putative thermoregulatory savings may be offset by the increased expense of migration itself or mediated by changes in the functional organ size and performance[22]. However, during several key periods of the non-breeding phase, migrant blackbirds displayed individualized metabolic dynamics apparently aimed at offsetting migration costs.

Starting 28 days before fall migration departure, future migrants nocturnally decreased $f_H$ relative to residents (EST of −12.67, SE of 7.57, $F = 1.96$; Fig 3a and Supplementary Table 2). This cumulative $f_H$

reduction, as evidenced by the mean across each strategy, amplified as departure approached (up to a maximum of −19.5% in beats per minute) and suggests substantial metabolic downregulations and energy-saving in advance of migration[23,24]. Migrants also concurrently reduced $T_b$ for 17 days before spring departure (EST of −0.04, SE of 0.03, $F = 1.96$; Fig. 3g and Supplementary Table 2). This suggests a potential mechanism for pre-departure energy conservation: migrants lower their $T_b$ setpoint[25], allowing nocturnal $T_b$ to decrease more in the lead-up-to-fall migration than in other phases. By reducing the energy expended on thermoregulation[26–28], migrants are able to allocate energy to other processes, such as fat accumulation for fuel storage[29,30] and the increase of flight muscles[31,32], both important components of preparation for migration. Differences in heart size between strategies (which would result in difference in stroke volume[33,34] and haematocrit values[35]) could, in principle, increase during the pre-migration phase, potentially decoupling $f_H$ from oxygen consumption and, thus, energy expenditure. However, our data show that the combined decreases in $f_H$ and $T_b$ specifically occured during nocturnal periods and were not observed during the day (EST of −0.07, SE of −0.08, $F = −12.29$; Supplementary Table 2 and Extended Data Fig. 1a,g). This suggests that the metabolic rate reduction is a strategic adaptation for night-time energy conservation rather than a general increase in heart efficiency. If heart size and stroke volume changes were primary factors, we would have also expected to see these effects during the day, which we did not.

Additionally, the near disappearance of this difference in the following spring (EST of −56, SE of 60.05, $F$ = 27.13; Fig. 3d and Supplementary Table 5) underscores the likelihood that the observed nocturnal $f_H$ and $T_b$ reductions are non-morphological pre-migratory adaptations, rather than changes in heart and cell physiology. Our findings show that the decision to migrate during fall precedes departure and requires physiological preparation well in advance of any movements, rather than an acute response triggered by environmental conditions, as had been previously suggested[36,37]. Similarly, in fall (30 days before the earliest migratory activity) and spring (after the arrival of all migrants), when all birds were in the shared breeding grounds, we observed no significant differences in overall $T_b$ or $f_H$, indicating comparable metabolic and thermoregulatory expenses in both strategies (Fig. 2a,g and Supplementary Table 1). We also observed no differences among strategies in the thermoregulatory response to changes in $T_a$ during fall season (Extended Data Fig. 2 and Supplementary Table 10). Together, these findings imply that the migratory strategy is not a simple function of individuals' inherent metabolic and thermoregulatory capabilities.

On nights with active migration, migrants exhibited a significantly higher $f_H$ compared with residents. Specifically, there was an increase of at least 25.9% (99 bpm) during fall migration and an even greater 36.4% increase (135 bpm) during spring migration (fall: EST of 98.75, SE of 11.29, $Z$ = 8.78, $P$ < 0.001, Fig. 3b and Supplementary Table 3; spring: EST of 135.30, SE of 9.60, $Z$ = 14.09, $P$ < 0.001, Fig. 3e and Supplementary Table 6). This increase in $f_H$ was complemented by a 0.7–0.9 °C elevation in $T_b$ for migrants (fall: EST of 0.97, SE of 0.08, $Z$ = 12.63, $P$ < 0.001, Fig. 3h and Supplementary Table 3; spring: EST of 0.70, SE of 0.06, $Z$ = 10.95, $P$ < 0.001, Fig. 3k and Supplementary Table 6). The energy costs of actual flight were probably even higher because blackbirds rarely migrated continuously through an entire night and actively travelled on average during only four nights (Supplementary Table 9). When considering periods of active travel only, migrants showed 53.2% (199 bpm) higher $f_H$ (EST of 199.41, SE of 14.46, $t$ = 13.79, $P$ < 0.001; Supplementary Table 8 and Extended Data Fig. 3c), accompanied by a 1.23 °C higher $T_b$ (EST of 1.23, SE of 10.07, $t$ = 18.71, $P$ < 0.001; Supplementary Table 8 and Extended Data Fig. 3d) compared with resident birds at the same time. Interestingly, migrants' nocturnal $T_b$ during active flight was intermediate between resting $T_b$ (for example, during sleep) and non-migratory diurnal $T_b$ (Extended Data Fig. 3b,d). We hypothesize that elevated $T_b$ arises from increased muscle activity during flight[38] rather than any adjustment to the 'normal' daily cycle of $T_b$ setpoint regulation[17].

Immediately upon departing the breeding grounds, migrants demonstrated thermoregulatory advantages that did not translate to detectable differences in $f_H$. On stopovers, migrants already exhibited slightly higher $T_b$ compared with their resident counterparts (EST of 0.22, SE of 0.08, $Z$ = 2.84, $P$ = 0.005; Fig. 3h and Supplementary Table 3), potentially due to milder conditions (Fig. 1b and Supplementary Table 9). After final arrival, we observed that the nocturnal $T_b$ of migrants remained more consistent, while $T_b$ of resident birds continued to decrease seasonally at the same time[17], resulting in the lower winter $T_b$ for residents (Fig. 3i). We found no significant differences in $f_H$ between the two groups during fall stopovers (EST of −15.37, SE of 10.89, $Z$ = −1.41, $P$ = 0.16; Fig. 3b and Supplementary Table 3), consistent with the patterns observed during winter (Fig. 2b and Fig. 2d).

Upon arrival at wintering sites, migrants exhibited temporarily lower $f_H$ for up to eight days (EST of −36.27, SE of 11.96, $F$ = 1.96; Fig. 3c and Supplementary Table 4), indicating a short recovery phase[39] following completion of fall migration. A similar tendency could be seen already during earlier stopovers; however, the effect was not statistically significant (EST of −15.37, SE of 10.89, $Z$ = −1.41, $P$ = 0.16; Supplementary Table 3). It should be noted that incorporating numerous consecutive stopover nights[40] (Supplementary Table 9) could diminish any potential signal of recovery periods after active flights (Fig. 3b).

In contrast to fall, we found little evidence of pre-migratory metabolic adjustments during the lead-up spring migration. Migrants did exhibit a lower $f_H$ 3 days before spring departure (EST of −10.64, SE of 7.29, $F$ = 1.96; Fig. 3d and Supplementary Table 5). However, the magnitude of this reduction was relatively modest (9%), in comparison with fall (Fig. 3a). Notably, we did not observe any evidence of the nocturnal thermoregulatory downregulation, as was the case during fall. Although the $T_b$ of the migrant birds was marginally lower than that of the resident birds 6 days before departure (EST of −0.06, SE of 0.03, $F$ = 1.96; Fig. 3j and Supplementary Table 5), this difference was attributed to the seasonally increasing $T_b$ of the resident birds during this period, probably caused by a change in $T_a$ at the more northern breeding site[17]. It is possible that spring $T_a$ was simply too high or that the preparation of the reproductive system[41,42] already started, which, in turn, did not allow downregulation of $T_b$ during the night as observed during the fall, suggesting intrinsic differences between spring and fall pre-migratory programmes.

The pronounced differences in pre-migratory metabolic dynamics between fall and spring migration suggest that these periods involve different metabolic preparations and mechanisms. This notion complements existing evidence suggesting different drivers and strategies employed between the two seasonal journeys (for example, variations in migration speed and the rationale for timely arrival during these seasons[43–45]). Furthermore, our findings have important implications for understanding the potential influence of environmental changes on the energy balance and, ultimately, wintering decisions of migratory species[10,12]. For example, environmental factors may differentially affect aspects of fall and spring migrations.

## Differences in thermoregulation costs

To estimate differences in energy spent on thermoregulation between the strategies, we parameterized a blackbird-specific individual-based biophysical model of endothermic thermoregulation using only observed $T_b$ and observed or interpolated $T_a$[46] (the model does not use $f_H$; Methods, Extended Data Fig. 4 and Supplementary Table 11). This model predicted that resident blackbirds in substantially colder winter environments incurred markedly higher metabolic costs of thermoregulation than migrants (Fig. 4) despite maintaining a lower $T_b$ (Fig. 2h). This finding was robust to substantial variation in assumptions about $T_a$, which could arise from uncertainty about wintering locations, micro-climatic buffering or behavioural compensation. Thus, migrants may realize a thermoregulatory benefit of higher $T_a$ during winter, which apparently does not extend to the overall metabolic rate. Instead, the warmer $T_a$ experienced by migrants could provide other organismal temperature-related benefits, such as a more reactive immune system[47] or greater predator avoidance capabilities[48]. It is important to note that our current model assumes no strategy-specific morphological differences between migrants and residents that would affect insulative capacity. If present, such differences could change the estimated difference in estimated thermoregulatory energy expense. Previous work has found no difference in flight-related morphology (wing aspect ratio and tail length) between migratory strategies[49] but whether internal morphology differs remains unstudied.

### Energy in the life history of migration

According to our metabolic simulation model, on average, residents expended 18,564 kJ more energy on thermoregulation than migrants, an approximately 1.75-fold difference in allocation despite equivalent total energy expenditures. Thus, assuming approximately equal total energy expended between migrants and residents, as implied by our $f_H$ results, these 'savings' represent a potential energy surplus available to migrant blackbirds.

A portion of energy saved on thermoregulation could potentially be used to offset any increased costs associated with migration. However, it is unlikely that the additional costs of undertaking migration rise to this magnitude for several reasons. First, the attenuation of pre-migratory $f_H$ and lowering of minimum $T_b$ setpoint are expected to

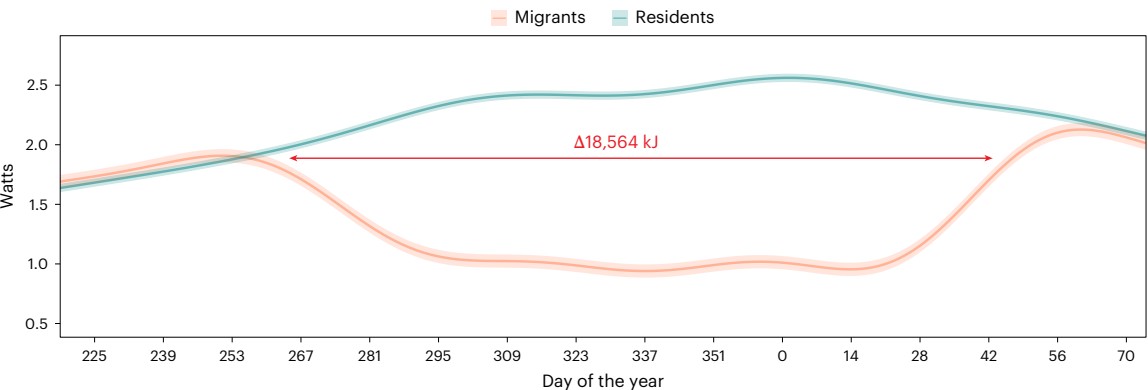

**Fig. 4 | Thermoregulatory simulation model.** Thermoregulatory simulation showing differential energy expenditures for migrant and resident blackbirds. The predicted energy expense of thermoregulation (lines) and 95% confidence intervals (ribbons) for both migrant and resident phenotypes derived from GAMM. Thermoregulatory metabolism is estimated on the basis of observed $T_a$ and $T_b$ (Methods). The periods, where confidence intervals do not overlap, indicate significantly different energetic expense of thermoregulation.

offset the net cost of fall and spring migration, at least to some extent. Second, the relatively limited number of active migration nights (range 1–9) during fall or spring migration (Supplementary Table 9) implies a comparatively minimal metabolic cost of migratory flights themselves, in line with findings in other thrush species[7]. In fact, we calculated that the energy cost of active migration constitutes only 6.7% (1.7–12.2%) of the estimated energy savings from reduced thermoregulatory needs in warmer environments[50] (Methods: 'Biophysical models', Supplementary Table 12). Instead, we propose that the observed differences in energy budgets may offer a novel explanation for the persistence of different wintering strategies, as their annual routines inevitably impose divergent pressures on individual fitness components.

Morphological differences which decouple $f_H$ from true metabolic rate could conceivably account for this discrepancy and would potentially reduce or eliminate this surplus. However, for that to be the case, resident birds would need to exhibit increased metabolic efficiency per heartbeat (that is, greater stroke volume) during winter. Seasonal changes in morphology and physiology, such as increased heart size and size of pectoralis muscles, have been observed in association with avian migration[22], as well as cold-induced thermogenesis[51]. Thus, future work comparing morphological plasticity, especially heart size, would be especially useful.

The pace of life theory[52] proposes that traditional life history trade-offs[53] are mediated by physiological mechanisms. The extended pace of life theory further suggests that behavioural strategies should predictably covary with life history strategies[54]. In the context of migration, understanding how a behavioural phenotype is linked to specific life history trade-offs has been hindered by the challenges in accurately estimating fitness components of migratory animals[55]. While not well estimated, preliminary findings suggest that survival rates may be lower during migrations[1] but may enhance or equalize survival probability during other seasons or on an annual basis[14,56]. Alternatively (or additionally), migrants may trade off survival costs with other fitness components, for example, by increasing fecundity[57]. Counterbalancing lower migration period survival with elevated survival during other periods and/or higher fecundity might be necessary to ensure equivalent long-term relative fitness among strategies. Interestingly, migratory blackbirds have previously been shown to exhibit higher annual survival than residents[14], but strategy-specific differences in, for example, clutch size remain unknown. Thus, migratory blackbirds may use the putative surplus energy arising from lower thermoregulatory burdens to better regulate body condition and, thus, reduce overall intrinsic mortality risk.

Previous hypotheses to explain the emergence and maintenance of partial migration focus on intraspecific competition, positing either a frequency-dependent evolutionary stable strategy, a conditional frequency-dependent strategy (fitness contingent on individual traits)[58,59] and/or density-dependent effects of seasonal resource fluctuations[60,61]. While competition-based theories do not exclude the possibility that individuals modulate fitness components to achieve equivalent overall fitness[62], these trade-offs are typically viewed as secondary consequences and not ultimate explanations. More recent work has framed the evolution of bird migration as an explicitly individual phenomenon to follow environmental seasonality and, in this way, escape from harsh winter conditions (rather than density-dependent optimal resource tracking)[3,63]. In this framing, winter residency in seasonal environments requires as much explanation as migration because overwintering residents must contend with the many challenges posed by winter[3]. Frequency- and/or density-dependent explanations are only partly satisfactory because they only consider changes in resource distributions and ignore other facets of environmental dynamism that accompany winter. Our findings present a competition-free, individual-based mechanism for modulating energy allocations across non-breeding strategies which could result in concurrent variance in fitness components.

Overall, we did not find differences in total energy spent between migrants and residents. However, different thermoregulatory contexts apparently drive varying energy budgets offset by other currently unknown costs. An individual choice of residency over migration suggests a bi-modal distribution of life history strategies in the species. The exact physiological mechanisms by which individuals 'choose' one strategy over the other still need to be determined, but our data at least help to solve the debate whether energy trade-offs are involved in such decisions. These insights emphasise the importance of incorporating field-based energetic measurements in the re-evaluation, refinement and potential rejection of long-standing dogma in the field.

## Methods

### Study area and captures
We captured a total of 118 adult common blackbirds (*Turdus merula*) in a mixed forest in southern Germany (47.7801° N, 9.0203° E) over three consecutive years (2016–2018). This population is partially migratory— about 26% of all individuals migrate in autumn (female 36%, male 16%)[11]. Adult blackbirds (both sexes) were caught with mist nets, fitted with an aluminium leg ring, and transported in cloth cages (height 30 cm, width 26.5 cm, length 49.5 cm) to the Max Planck Institute of Animal Behaviour, Radolfzell (~10 min drive). At the institute, we surgically implanted internal $f_H$ and $T_b$ loggers ('Surgery' section) and affixed external radio transmitters. Birds were then returned to their original capture location and released.

## Surgery

We placed birds on a 40 °C heating pad to prevent hypothermia and then anaesthetized them with isoflurane (CPH Pharma CP 1 ml ml⁻¹, %5). We continuously controlled the bird's $T_b$ and monitored its breathing frequency. We injected 2 ml of ringer solution into the femur tibia joint to avoid extensive dehydration. After carefully removing some abdominal feathers, we made a 10 mm abdominal incision in the skin and tissue layer beneath the sternum. Star-Oddi DST micro-HRT/temperature data loggers (Star-Oddi, dimensions 8.3 mm × 25.4 mm, weight 3.3 g), which were gas sterilized with ethylene-oxide at 38 °C before (done at Osypka AG), were inserted after which skin and tissue were stitched separately with an absorbent suture. We then monitored the recovery of the bird in hand and, after ensuring its well-being and normal behaviour, we attached a backpack with a radio transmitter (≤2.6 g; produced by Sparrow Systems, the Swiss Ornithological Institute or Holohil Systems) via a leg-loop harness to the bird. The mean weight of a blackbird is about 90 g. Thus, external radio tags, in combination with an implanted $f_H$ and $T_b$ logger, add approximately 6.56% (5.9 g) to total body mass. The weight of the transmitter varied from -1.8 g to 2.6 g, with heavier birds receiving the heavier tags to mitigate the relative burden. Besides the weight, the aerodynamic effects of external tags could notably impact bird activities. Due to their location within the body, the implanted loggers probably have reduced aerodynamic influence, contributing to lesser negative impacts on the birds. To provide some recovery time after surgery and to transport the bird back to the catching site, we placed birds back in a cloth cage where water and food were available ad libitum. In 2015, we conducted a pilot study with five blackbirds kept in aviaries to test their response to implanted loggers. We verified the physical health after this type of surgery and observed that wound healing was not affected after a short recovery phase. Furthermore, during the main study, recapture and migratory return rates were not lower for birds with implanted loggers compared with only radiotagged birds from previous years. The return rates for birds with implanted loggers were 90%, compared with 43% for the control group, and recapture rates were 80% versus 23%, respectively. The experimental setup may have influenced these findings, which required extensive recapturing efforts to retrieve the loggers and continuous monitoring, making direct comparisons challenging.

## Data collection

The attached radio transmitter backpacks enabled us to determine the status (presence/absence and alive/dead), non-breeding strategy (migrant versus resident) and the timing of departures and arrivals of individuals at the breeding site. To this end, we deployed six automated receiver units (ARU, Sparrow Systems) at selected locations in the study site[64], where each ARU searched for up to 60 frequencies chosen within a maximum time frame of 240 s. The ARUs were connected to H antennas, mounted at 3–12 m. A total of 24 h ARU monitoring allowed us to precisely determine departure and arrival events via an initial rapid increase in the signal strength of the radio transmitters, followed by a steady decline during fall or a sudden reappearance accompanied by an increase in signal strength and continuous presence afterwards. We later used visual controls of ARU data sightings and manual handheld tracking to ensure the absence or presence of an individual within a 2.5 km radius. Manual tracking was performed using a handheld H antenna (Andreas Wagener Telemetry Systems) and a Yaesu VR 500 receiver (Vertex Standard USA). We also used car-mounted Yagi-antennas (AF Antronics) and an airplane equipped with two H antennas and two Biotrack receivers (Lotek) to ensure the departure of an individual within a 20 km radius of the study site. All post-breeding departures between 2 September and 24 November were included in our analysis. Later departures were classified as 'winter migration' or irruptive migration[11] and excluded from this study.

The implanted data loggers were programed to start recording on 1 September at 1:00. They recorded $f_H$ at 600 Hz and core $T_b$ every 30 min, including a measure of the signal-to-noise ratio (quality index, QI) of the electrocardiogram (ECG). Additionally, raw ECG measurements were saved every 60 h for later verification of data quality ('Pre-processing of $f_H$ and $T_b$ data' section).

## Recapture

We attempted to recapture all birds for data extraction during the following spring. We used the telemetry-derived positions of the birds (either on-site throughout the winter or whose return was recorded by the ARUs) to precisely target recapture using mist nets. After surgical extraction of the data loggers (using the same protocols as for implantation), the birds were released at their capture site. The data on the loggers were downloaded using the Mercury program (Star-Oddi).

## Sample sizes

We implanted 118 loggers from 2016 to 2018 and were able to recapture 83 birds from 2017 to 2019. From that, we get a total of 890,689 measurements (see Supplementary Table 1 for the exact distribution of the measurements).

## Pre-processing of $f_H$ and $T_b$ data

Although $T_b$ measurements were pre-calibrated to ±0.2 °C during production, the quality of the collected $f_H$ measurements depends on the individual-specific signal-to-noise ratio and varies considerably between the loggers. Since the QI, a measure of the signal-to-noise ratio provided by the logger algorithm is based on all previously taken measurements in each logger, it is not comparable between loggers and therefore requires individual filtering. We used the raw ECG data saved every 60 h to include only reliable measurements with known uncertainty. We manually calculated the correct beats per minute for these measurements via the raw ECG trace plots and compared this with the one internally calculated by the logger algorithm. We then individually estimated the assigned error rate for each logger and QI's. We filtered all data accordingly to include only the QI with a known error rate.

Furthermore, a manual calculation of all ECGs allowed us to determine the maximum and minimum plausible $f_H$ that can be observed and verified in the field. After final filtering, we excluded 12 loggers due to insufficient data quality. We expected only measurements with a QI error rate of less than 15% and exhibiting values within the known range of reasonable $f_H$. The final data set for analysis included 510,654 and 597,321 measurements of $f_H$ and $T_b$, respectively.

## Classification of migration

We used the known breeding site departure and arrival dates for migratory birds recorded via ARU radio telemetry[11] to train a gradient-boosted machine-learning model (R package 'gbm'[65]) based on $f_H$, $T_b$, individual logger identification, individually scaled temperature and $f_H$, the difference to the mean $f_H$ and $T_b$ and proportional temperature increase. The model classified all nightly measurements between departure and arrival as migration or stationary phases. Afterwards, we visually classified all measurements by ourselves and compared our manual classification with the one via the machine-learning model. Both classifications matched by 97.7% (model building AUC 0.966, classification AUC 0.977). We then used these data to predict arrival on and departure from the wintering sites as well as stopover periods, which were not observable via ARU radio telemetry.

## Definition of seasons and individual key migratory stages

In addition to comparing the $f_H$ and $T_b$ of the two migratory phenotypes, we also defined three main calendar seasons for a more focused analysis.

We defined the first 7 days of measurement (1–7 September) as fall, where all individuals of both strategies are in the same location, have finished breeding but are still relatively far away (30 days) from the first recorded departure of a migratory blackbird (on 11 October). We conservatively defined winter as the 46 days between the last fall

and first spring migration events detected for our blackbirds (3 December until 17 January). During this time, birds of the two overwintering phenotypes are spatially separated and reside at their respective wintering sites.

The arrival of the last migratory blackbird at the breeding site (2 April) marks the start of our definition of the post-migration spring season. It spans 8 days until April 10, when the sample size of migratory birds becomes less than five, owing to recapture and battery depletion.

Because we observed high individual variance in the phenology of migratory events (for example, departure and arrival timing, duration and so on (Fig. 2d,f)), for some analyses, we standardized $T_b$ and $f_H$ on the migration-relevant transition events (rather than calendar dates) for eight stages of the life cycle. The first period is the fall pre-migration phase (35 days before fall departure), followed by fall migration and stopover periods, which mark the time between initial departure and last arrival before the core winter season starts. The very last fall migration starts the winter arrival (first 14 days after arrival in the wintering site), which turns into the core winter from the calendar-based analysis (Fig. 2d,f, 'Winter' area). The following year, the return migration period starts with a spring pre-migration phase (21 days before spring departure), followed by spring migration, spring stopover and finally, spring arrival (first 14 days after arriving back at the breeding site).

Previous work shows that physiological responses to environmental conditions and seasonal adaptations can differ day and night. As blackbirds, to the best of our knowledge, migrate only at night, we also separated the analysis for day and night (Fig. 3 and Extended Data Fig. 1)

### Weather data
For Fig. 1b,c, we obtained monthly mean temperature data in a 2.5 min spatial resolution from the 'WorldClim' dataset (R package 'geodata'[66]). The temperature data of our study population's 25 known wintering areas based on previous tracking using geolocators of the same population[12] were annotated using the Env-DATA System on Movebank[67]. We used the hourly 'ECMWF ERA5 SL' temperature (2 m above ground) accounting for atmospheric conditions and the inverse distance weighted between the weather stations. To compare the conditions between breeding and wintering areas, the annual average and the corresponding 25% and 75% quantiles were calculated in December and February, as these are the general periods when both phenotypes (migrants and residents) are spatially separated in their respective wintering areas. Since the environmental data were available at hourly resolution, but physiological measurements were taken every half hour, we linearly interpolated the $T_a$. We used these extracted $T_a$ to estimate strategy-specific thermoregulatory energy expenditures ('Biophysical models' section). To assign the estimated $T_a$ to the respective migratory birds based on their progress towards their wintering grounds, we divided the migration period for each bird by the number of migration nights it undertook, segmenting the journey accordingly. With each migratory night, the experienced $T_a$ then converges linearly towards the respective temperature mean of the wintering sites or the 25/75% quantile. During spring migration, the temperatures experienced gradually adjusted to the temperatures of the breeding site in the same way.

### Statistics
To test for differences in $f_H$ and $T_b$ for resident and migratory blackbirds in different calendar periods, we used a linear mixed model (R package 'lme4'[68]) with individual measurements of $f_H/T_b$ on a resolution of 30 min as a response variable and wintering strategy, calendar season, day phase and sex as predictors. The birds' identification and date were included as random factors.

To analyse energetic differences at various migration stages ('Definition of seasons and individual key migratory stages' section), we assigned each single $f_H$ and $T_b$ measurement of resident birds to simultaneous single measurements of migratory individuals of the same sex based on the real-time timestamp. By distributing all

measurements of resident birds ($N = 54$) from the same sex equally among the migratory birds ($N = 19$), every single measurement from a resident was only referenced once to a specific measurement of a migrant bird. This assigned each single measurement of a resident a 'stage of migration', corresponding to the reference migrant measurement, allowing us to directly compare the physiological data of residents and migratory blackbirds in relation to the departure and arrival events of the migrants. Since each resident measurement was assigned only once, the dataset contains unique occurrences of each measurement, thereby avoiding any pseudoreplication. We performed migration stage-centred analysis with generalized additive mixed models (GAMM, R package 'mgcv'[69]), including $f_H/T_b$ measurements again as the response variable. Each migration stage was analysed in a separate model, and the days before and after arrival and departure events have been used as a smoothing factor. Wintering strategy and sex were included as predictors. In both analyses, we eliminated temporal autocorrelation, following the established procedure of randomly discarding 30% of the data from each individual[17,70]. In addition, the birds' identification and date were included as random factors to account for individual-specific variation and repeated measurements. We used a post hoc test with a Bonferroni correction to calculate pairwise comparisons in each season.

### Biophysical models
To estimate strategy-specific thermoregulatory energy expenditures, we used an instantiation of the endotherm model contained in the 'nichemapr' package[46]. This model, based on Porter and Kearny (2009)[71], estimates the dynamic metabolic expenditure of a homoeostatic endotherm based on taxon-specific morphological parameters and typified behavioural responses to thermal fluctuations. The performance of this model has been widely validated against empirical measurement, including in birds[72–75]. We fitted models based on observed and interpolated $T_a$ ('Weather data' section) and the bio-logger-recorded $T_b$. Species-specific functional trait values can be found in Supplementary Table 11. Thus, we produced dynamic metabolic models for thermoregulation for all 73 individual blackbirds in our dataset. To capture potential uncertainty in $T_a$ during winter for both residents and migrants, we considered alternative $T_a$ timeseries for each. It is possible that resident individuals' experienced $T_a$ was slightly higher than weather-station observations due to micro-climatic buffering. Thus, we considered scenarios wherein we added 1 °C and 2 °C to the observed temperatures for the resident birds during the period when migrants were off-site (the most conservative possible difference). Similarly, because over-winter $T_a$ was estimated from geolocator-based estimates of winter range from previously studied birds of the same population[14], we also considered the minimum and maximum temperatures possible within the migrants' possible range to bracket the warmest and coldest possible $T_a$ timeseries. We compared all combinations of these scenarios to evaluate the sensitivity of our results to the specific temperature timeseries.

To quantify the differences in the energy expense of thermoregulation between migrants and residents, we fit a hierarchical GAMM for thermoregulatory expenditure (output from the 'nichemapr' model) as a function of Julian day interacted with migratory strategy using the 'mgcv' package in R (ref. 69). We used a thin plate smoothing term and included a random intercept by individual year to account for individual differences in metabolic rate (for example, body size variation). This allowed us to directly model thermoregulatory metabolic expense as an individual-based timeseries dependent on migratory strategy.

### Energy expenditure of migratory flights
To estimate the energy expenditure for individual migratory journeys, we applied an allometric equation derived from Bishop and Butler (2015)[50], $y = 52.6M^{0.74}$, where $y$ represents the power required for flight in watts ($J s^{-1}$) and $M$ is the body mass in kilograms.

Using this equation, we calculated the power required for each bird's flight and multiplied the power by the total flight duration in seconds to obtain the total energy expenditure in joules (Supplementary Table 12).

## Reporting summary

Further information on research design is available in the Nature Portfolio Reporting Summary linked to this article.

## Data availability

The datasets supporting the conclusions of this article are available in the figshare data repository at https://doi.org/10.6084/m9.figshare.24799596.

## Code availability

The code for the biophysical models in this article is available as a GitHub repository at https://github.com/syanco/blackbird_metabolics. All other analyses used standard software and scripts as described in Methods and Supplementary Information.

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

## Acknowledgements

We thank A. Schmidt, A. Meltzer, M. Borho, L. Kettemer and D. Hägele for their help as field assistants in collecting the data. L. Keicher performed surgeries. Á. Bjarnason helped understand the data and developed loggers. Y. Pei provided the blackbird artwork in Fig. 1a. We also thank D. Dent, J. W. Arnold, R. Shipley, K. Morelle and A. Koelzsch for discussions and comments on the manuscript. This work was funded by the Max Planck Gesellschaft and was approved by the responsible ethic commission and ministry in Germany: Regierungspräsidium Freiburg, 35-9185.81/G-16/115, 35-9185.81/G-13/29 and 35-9185.81/G-09/08 (receiving authors: N.L., T.V., D.Z., M.W. and J.P.).

## Author contributions

Conceptualization: N.L. and J.P. Methodology: N.L., S.W.Y., D.Z., M.W. and J.P. Investigation: N.L., S.W.Y., T.V., M.W. and J.P. Data curation: N.L. Formal analysis: N.L. and S.W.Y. Visualization: N.L. and S.W.Y. Funding acquisition: M.W. and J.P. Project administration: N.L. and J.P. Supervision: J.P. Writing—original draft: N.L., S.W.Y. and J.P. Writing—review and editing: N.L., S.W.Y., T.V., M.W. and J.P.

## Funding

## Competing interests

The authors declare no competing interests.

## Additional information

**Extended data** is available for this paper at https://doi.org/10.1038/s41559-024-02545-y.

**Correspondence and requests for materials** should be addressed to Nils Linek.

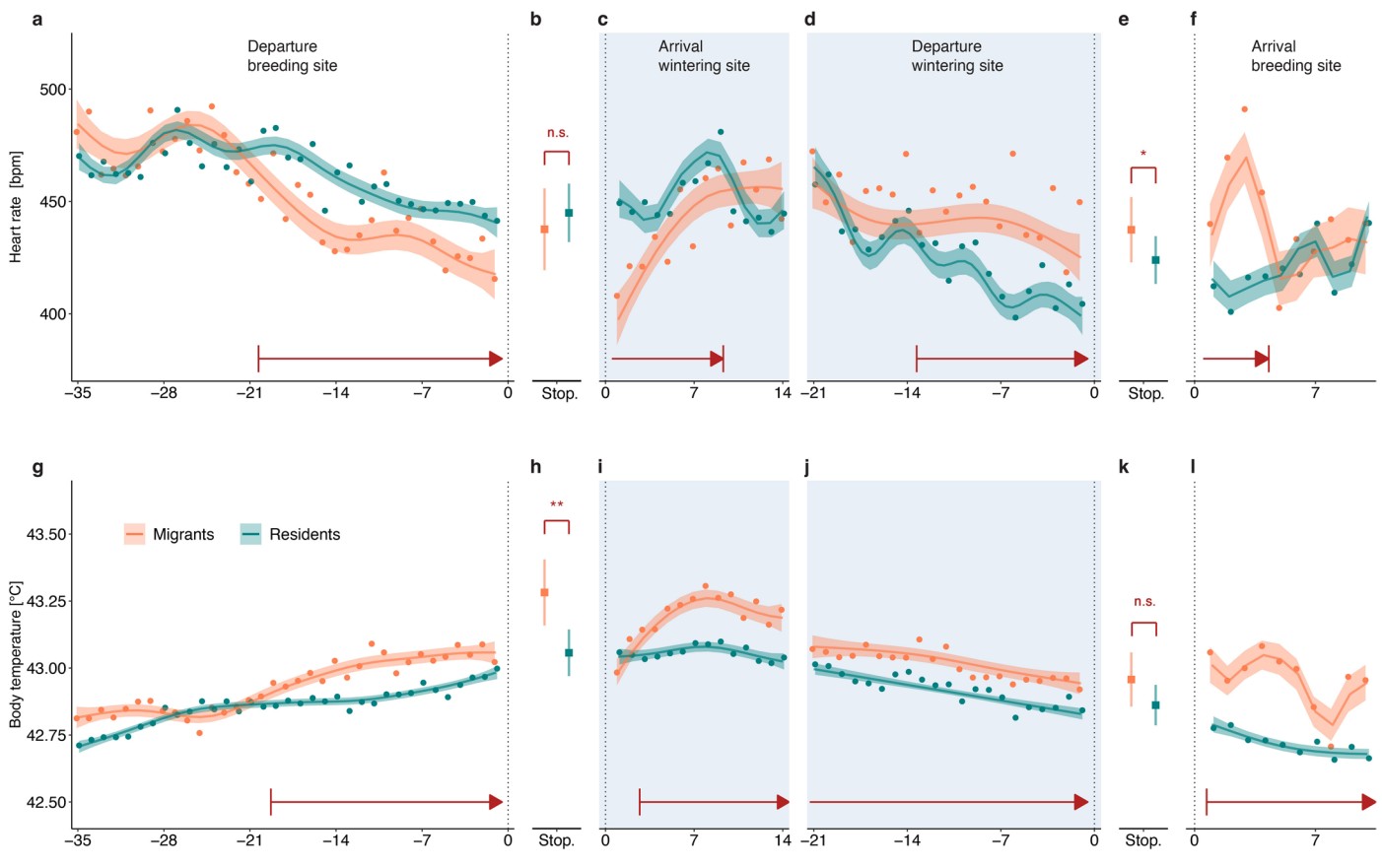

**Extended Data Fig. 1 | Comparison of heart rate and body temperature between strategies relative to migration stages during day. a-f**, Mean heart rate and **g-l**, body temperature are shown as points during day across all migrants (orange) and all residents (dark green) centred on departure date from breeding site relative to initial departure **(a,g)**, stopover **(b,h)** in fall, centred on arrival date in wintering site **(c,i)**, centred on departure date from wintering site relative to spring departure **(d,j)**, stopover in spring **(e,k)**, centred on arrival date on breeding site **(f,l)**. For all measurements over time **(a,c,d,f,g,i,j,k,l)**, each single point represents the mean value across each overwintering strategy, with migrants centred and residents correspondingly assigned. The coloured solid line shows predicted heart rate and body temperature values for each strategy derived from a generalised additive mixed model, including individual measurements for each bird. Correspondingly coloured ribbons show the 95% confidence interval of those predictions. Blue-marked periods highlight the time when migratory birds reside in their final wintering grounds. Horizontal red arrows mark the first and last times when measures significantly differ between strategies. For migration stage-centered comparisons via linear mixed models **(b,e,h,k)**, means are shown as colored squares with standard error bars (SEM). Bonferroni corrected statistical significance levels: *** = $p < 0.001$, ** = $p < 0.01$, * = $p < 0.05$, and 'n.s.' = $p > 0.05$.

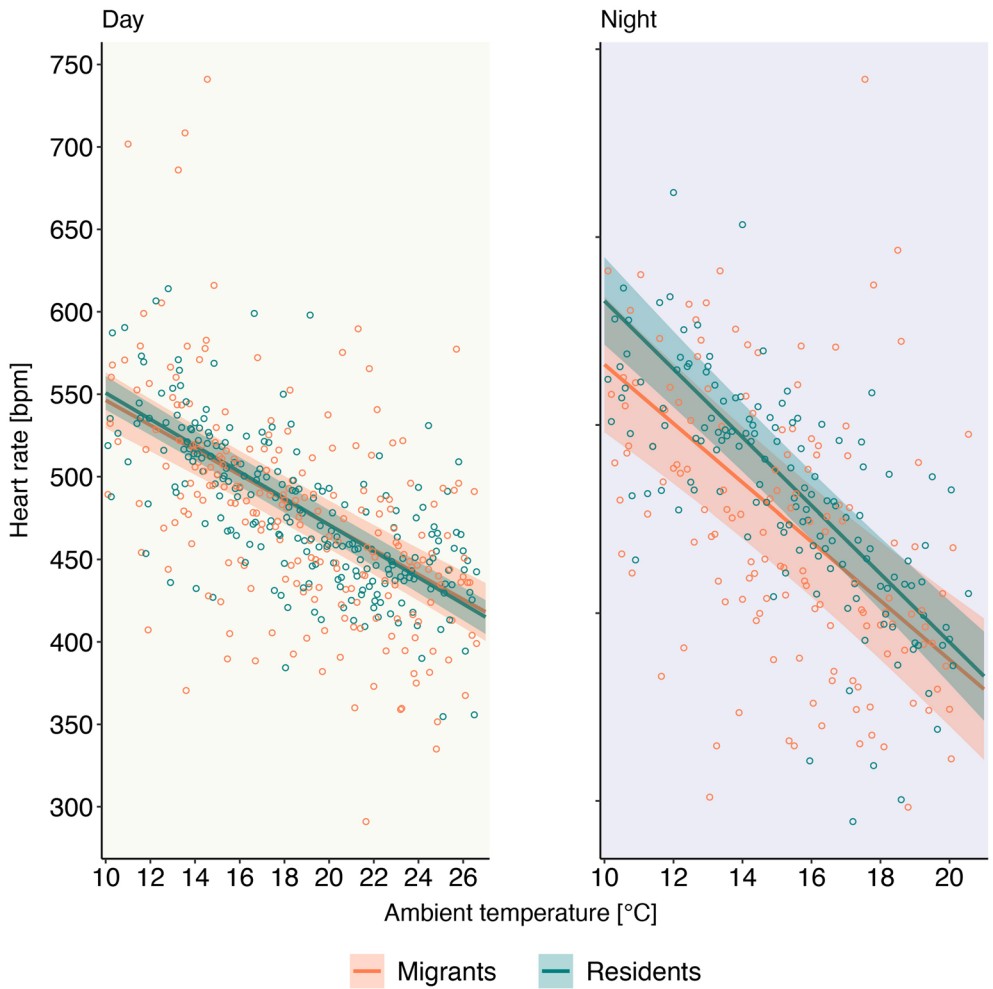

**Extended Data Fig. 2 | Visualization of reactions in HRT to different ambient temperatures for both wintering strategies.** Mean heart rate of resident and migratory blackbirds in relation to ambient temperature during day and night. Plotted circles are $f_H$ mean values for all occurring temperatures during fall (1st Sep.–7th Sep.). Lines are predicted values of the calculated linear mixed model (Supplementary Table 10) with respective 95% confidence intervals as ribbons around them.

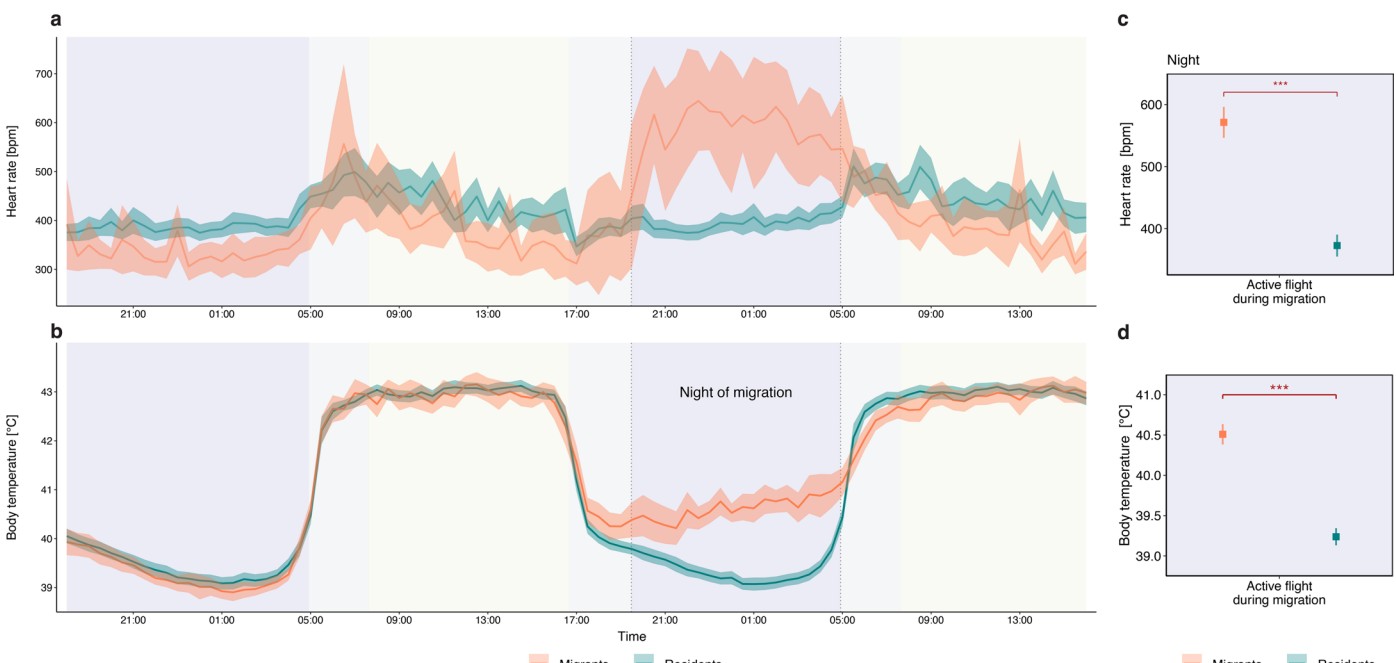

**Extended Data Fig. 3 | Heart rate and body temperature relative to the initial migratory departure. a**, Mean heart rate and **b**, body temperature in 30-minute intervals over 2 consecutive hours with 95% confidence interval bands for migrants and residents relative to the initial migratory departure of the migrants. Data of resident birds have been individually aligned to birds of the same sex and at the same date and time. The light blue periods mark night-time for all fall migrants, orange periods mark daytime, and light grey periods in between are estimated dusk and dawn phases, depending on the exact departure date.

**c**, Comparison of mean predicted heart rate and (**d**) mean body temperature via linear mixed model between migrants and residents with 95% confidence interval bars. Significant differences, derived from a linear mixed model and Bonferroni corrected (see Supplementary Data Table 1 and Supplementary Results) are indicated by asterisks: *** = p < 0.001, ** = p < 0.01, * = p < 0.05, and 'n.s.' = p > 0.05. Analysed data include only active flight periods during migration nights from initial departure up to final arrival returning at the breeding site.

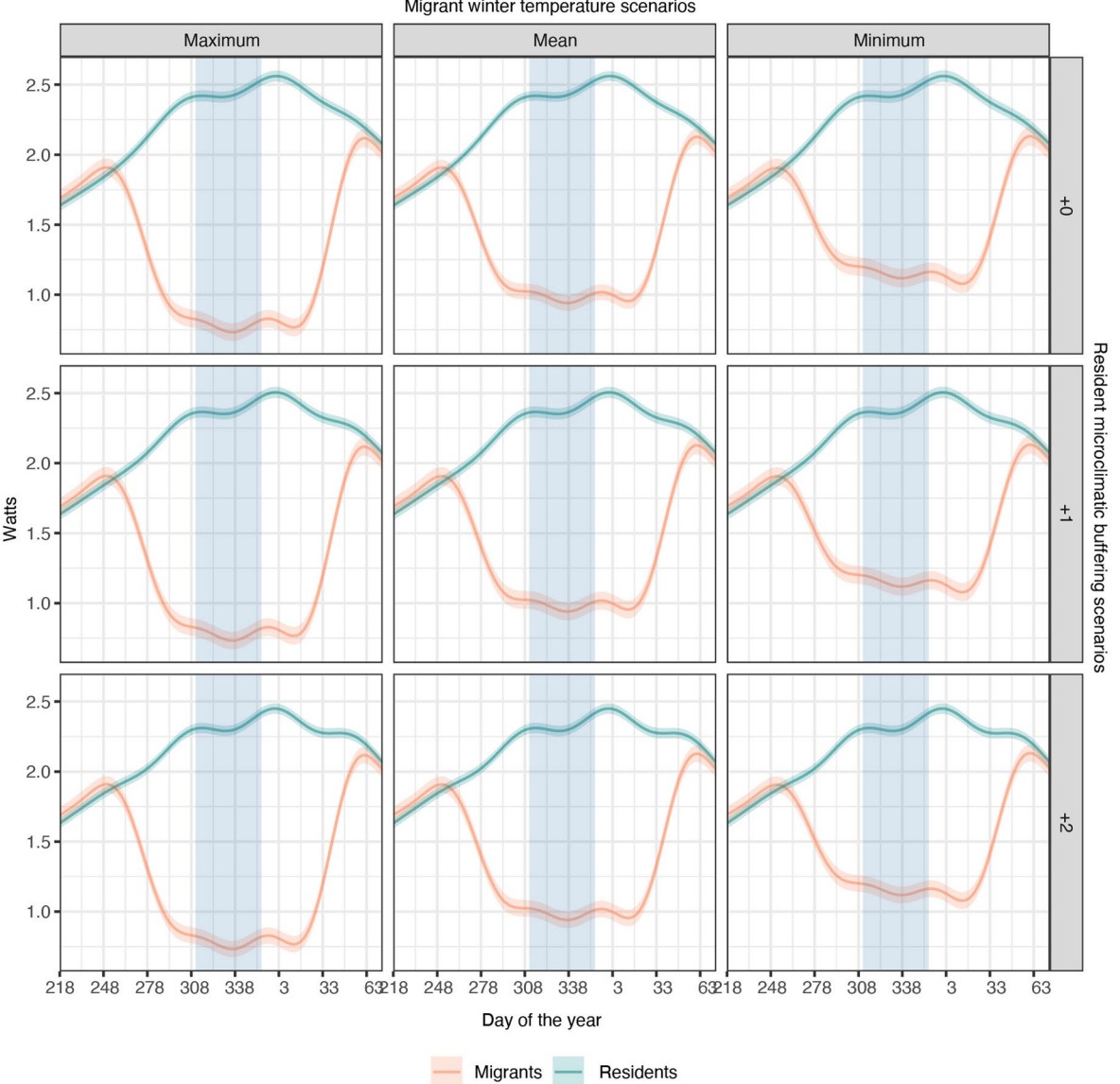

**Extended Data Fig. 4 | Alternative thermoregulatory scenarios.** Energetic expenditures on thermoregulation over time across migrant and resident blackbirds. To quantify the sensitivity of our findings to alternative $T_a$ timeseries, we considered alternatives for both overwintering residents as well as migrants. For migrants we considered $T_a$ timeseries comprised of the mean $T_a$ across the winter range (top middle, primary result in main text) but also considered 25% and 75% temperature quantiles from across the range on each day. On the breeding grounds, the ambient temperature is better estimated but does not include the potential for buffering via the disproportionate use of warmer micro-climates. Thus, we considered two extreme alternative scenarios wherein we inflated the $T_a$ for wintering residents (but not for migrants) by one and two degrees (rows). Blue shaded area denotes the core winter period.

# Reporting Summary

## Statistics

For all statistical analyses, confirm that the following items are present in the figure legend, table legend, main text, or Methods section.

| n/a | Confirmed | |
|---|---|---|
| ☐ | ☒ | The exact sample size (*n*) for each experimental group/condition, given as a discrete number and unit of measurement |
| ☐ | ☒ | A statement on whether measurements were taken from distinct samples or whether the same sample was measured repeatedly |
| ☐ | ☒ | The statistical test(s) used AND whether they are one- or two-sided<br>*Only common tests should be described solely by name; describe more complex techniques in the Methods section.* |
| ☐ | ☒ | A description of all covariates tested |
| ☐ | ☒ | A description of any assumptions or corrections, such as tests of normality and adjustment for multiple comparisons |
| ☐ | ☒ | A full description of the statistical parameters including central tendency (e.g. means) or other basic estimates (e.g. regression coefficient) AND variation (e.g. standard deviation) or associated estimates of uncertainty (e.g. confidence intervals) |
| ☐ | ☒ | For null hypothesis testing, the test statistic (e.g. *F*, *t*, *r*) with confidence intervals, effect sizes, degrees of freedom and *P* value noted<br>*Give P values as exact values whenever suitable.* |
| ☒ | ☐ | For Bayesian analysis, information on the choice of priors and Markov chain Monte Carlo settings |
| ☒ | ☐ | For hierarchical and complex designs, identification of the appropriate level for tests and full reporting of outcomes |
| ☐ | ☒ | Estimates of effect sizes (e.g. Cohen's *d*, Pearson's *r*), indicating how they were calculated |

*Our web collection on statistics for biologists contains articles on many of the points above.*

## Software and code

Policy information about availability of computer code

| Data collection | For logger setup the StarOddi Mercury Software (6.41) was used. |
|---|---|
| Data analysis | R 4.2.1<br>rstatix 0.7.2<br>data.table 1.14.8<br>scales 1.2.1<br>socviz 1.2<br>lubridate 1.9.3<br>dplyr 1.1.3<br>plyr 1.8.9<br>suncalc 0.5.1<br>lme4 1.1.35.1<br>multcomp 1.4.25<br>visreg 2.7.0<br>lmerTest 3.1.3<br>sjPlot 2.8.15<br>mgcv 1.9.0<br>itsadug 2.4.1<br>car 3.1.2<br>gdata 3.0.0<br>RcmdrMisc 2.9.1 |

emmeans 1.8.9
readr 2.1.4
binr 1.1.1
lsr 0.5.2
sjlabelled 1.2.0
sjmisc 2.8.9
sjstats 0.18.2
climwin 1.2.3
MuMIn 1.47.5
lattice 0.22.5
gmodels 2.18.1.1
lemon 0.4.7
terra 1.7.55
tidyterra 0.4.0
sf 1.0.14
rnaturalearth 0.3.4
tidyverse 2.0.0
Cairo 1.6.1
ggdark 0.2.1
rnaturalearthdata 0.1.0
maptools 1.1.8
ggmap 3.0.2
maps 3.4.1.1
rworldmap 1.3.8
rgdal 1.6.7
memisc 0.99.31.6
assertthat 0.2.1
sqldf 0.4.11
magrittr 2.0.3
reshape2 1.4.4
oz 1.0.22
scatterpie 0.2.1
Rmisc 1.5.1
zoo 1.8.12
signal 0.7.7
oce 1.8.1
gam 1.22.2
doParallel 1.0.17
doMC 1.3.8
MASS 7.3.60
hrbrthemes 0.8.0
viridis 0.6.4
ggdist 3.3.0
ggpp 0.5.5

For manuscripts utilizing custom algorithms or software that are central to the research but not yet described in published literature, software must be made available to editors and reviewers. We strongly encourage code deposition in a community repository (e.g. GitHub). See the Nature Portfolio guidelines for submitting code & software for further information.

# Data

Policy information about availability of data

All manuscripts must include a data availability statement. This statement should provide the following information, where applicable:
- Accession codes, unique identifiers, or web links for publicly available datasets
- A description of any restrictions on data availability
- For clinical datasets or third party data, please ensure that the statement adheres to our policy

The datasets supporting the conclusions of this article are available in the Dryad data repository (https://doi.org/10.5061/dryad.z612jm6jj).

# Research involving human participants, their data, or biological material

Policy information about studies with human participants or human data. See also policy information about sex, gender (identity/presentation), and sexual orientation and race, ethnicity and racism.

| Reporting on sex and gender | *Use the terms sex (biological attribute) and gender (shaped by social and cultural circumstances) carefully in order to avoid confusing both terms. Indicate if findings apply to only one sex or gender; describe whether sex and gender were considered in study design; whether sex and/or gender was determined based on self-reporting or assigned and methods used.*<br>*Provide in the source data disaggregated sex and gender data, where this information has been collected, and if consent has been obtained for sharing of individual-level data; provide overall numbers in this Reporting Summary. Please state if this information has not been collected.*<br>*Report sex- and gender-based analyses where performed, justify reasons for lack of sex- and gender-based analysis.* |
|---|---|

| Reporting on race, ethnicity, or other socially relevant groupings | *Please specify the socially constructed or socially relevant categorization variable(s) used in your manuscript and explain why they were used. Please note that such variables should not be used as proxies for other socially constructed/relevant variables (for example, race or ethnicity should not be used as a proxy for socioeconomic status).*<br>*Provide clear definitions of the relevant terms used, how they were provided (by the participants/respondents, the researchers, or third parties), and the method(s) used to classify people into the different categories (e.g. self-report, census or administrative data, social media data, etc.)*<br>*Please provide details about how you controlled for confounding variables in your analyses.* |
|---|---|
| Population characteristics | *Describe the covariate-relevant population characteristics of the human research participants (e.g. age, genotypic information, past and current diagnosis and treatment categories). If you filled out the behavioural & social sciences study design questions and have nothing to add here, write "See above."* |
| Recruitment | *Describe how participants were recruited. Outline any potential self-selection bias or other biases that may be present and how these are likely to impact results.* |
| Ethics oversight | *Identify the organization(s) that approved the study protocol.* |

Note that full information on the approval of the study protocol must also be provided in the manuscript.

# Field-specific reporting

Please select the one below that is the best fit for your research. If you are not sure, read the appropriate sections before making your selection.

☐ Life sciences  ☐ Behavioural & social sciences  ☒ Ecological, evolutionary & environmental sciences

For a reference copy of the document with all sections, see nature.com/documents/nr-reporting-summary-flat.pdf

# Ecological, evolutionary & environmental sciences study design

All studies must disclose on these points even when the disclosure is negative.

| Study description | The study compares migratory and resident blackbirds in southern Germany, focusing on heart rate as a proxy for energy expenditure. The design includes surgically implanted bio-loggers in individual birds to measure heart rate and body temperature in combination with classical radio transmitter backpacks. This approach allows for assessing variations in metabolic and thermoregulatory dynamics among different overwintering phenotypes and phases. The study aims to determine whether migration incurs additional energy costs and how energy is allocated across various overwintering stages. |
|---|---|
| Research sample | The research sample comprises partially migratory common blackbirds (Turdus merula) from southern Germany. The sample includes both migratory and resident adult birds, selected to represent the broader population of blackbirds in this region. Male and female birds have been equally included and were chosen to provide insights into differences in energy expenditure between migratory and resident phenotypes. |
| Sampling strategy | Since the overwintering strategy of each individual common blackbird was unknown at the point of capture, a sufficiently large sample size was necessary. This approach ensured adequate representation of both migratory and resident birds. Previous studies indicated that approximately 25% of the population migrates, guiding the decision on the number of birds to include in the study to capture a representative sample of both behavioral strategies. |
| Data collection | The attached radio transmitter backpacks enabled us to determine the status (presence/absence and alive/dead), nonbreeding strategy (migrant versus resident), and the timing of departures of individuals. To this end, we deployed six automated receiver units (ARU, Sparrow Systems, Fisher, IL, USA) at selected locations in the study site 51, where each ARU searched for up to 60 frequencies chosen within a maximum time frame of 240 seconds. The ARUs were connected to H antennas (ATS, Isanti, MN, USA), mounted at 3 to 12 m. 24-hour ARU monitoring allowed us to precisely determine departure and arrival events via an initial rapid increase in the signal strength of the radio transmitters, followed by a steady decline. We later used visual controls of ARU data sightings and manual handheld tracking to ensure the absence of an individual within a 2.5 km radius. Manual tracking was performed using a handheld H antenna (Andreas Wagener Telemetry Systems, Köln, DE) and a Yaesu VR 500 receiver (Vertex Standard USA, Cypress, CA, USA). We also used car-mounted Yagi-antennas (AF Antronics, Inc., Urbana, IL, USA) and an airplane equipped with two H-antennas and two Biotrack receivers (Lotek, Newmarket, ON, Can) to ensure the departure of an individual within a 20 km radius of the study site. All post-breeding departures between the 2nd of September and the 24th of November were included in our analysis. Later departures were classified as 'winter migration' or irruptive migration 11 and excluded from this study.<br><br>The implanted data loggers were programmed to start recording on September 1 at 1:00 a.m. They recorded heart rate (fH) at 600 Hz and core body temperature (Tb) every 30 minutes, including a measure of the signal-to-noise ratio (quality index) of the electrocardiogram (ECG). Additionally, raw ECG measurements were saved every 60 hours for later verification of data quality (see pre-processing of the fH and Tb data). |
| Timing and spatial scale | Season definitions and individual migration schedules<br>In addition to comparing the heart rates and body temperatures of the two migratory phenotypes, we also defined three main calendar seasons for a more focused analysis.<br>We defined the first seven days of measurement (1st Sep.-7th Sept) as fall, where all individuals of both strategies are in the same location, have finished breeding but are still relatively far away (30 days) from the first recorded departure of a migratory blackbird (on the 11th of October). We conservatively defined winter as the 46 days between the last fall- and first spring migration events |

detected for our blackbirds (3rd December till 17th Jan.). During this time, birds of the two migratory phenotypes are spatially separated and reside at their respective wintering sites.

The arrival of the last migratory blackbird at the breeding site (2nd April) marks the start of our define of the post-migration spring season. It spans eight days until April 10th, when the sample size of migratory birds becomes less than five owing to recapture and battery depletion.

Because we observed high individual variance in the phenology of migratory events (e.g., departure and arrival timing, duration, etc. (Fig. 2a and Fig. 2b), for some analyses we standardized Tb and fH on the migration-relevant transition events (rather than calendar dates) for nine stages of the life cycle. The first period is the fall pre-migration phase (35 days before fall departure), followed by fall migration and stopover periods, which mark the time between initial departure and last arrival before the core winter season starts. The very last fall migration starts the winter arrival (first 14 days after arrival in the wintering site), which turns into the core winter from the calendar-based analysis (see previous section). In the following year, the return migration period starts with a spring pre-migration phase (21 days before spring departure), followed by spring migration, spring stopover and finally, spring arrival (first 14 days after arriving back at the breeding site).

Previous work shows that physiological responses to environmental conditions and seasonal adaptations can differ day and night. Because blackbirds, to the best of our knowledge, migrate only at night, we also separated the analysis for day and night (Fig. 3, Extended Data Fig. 2)

| Data exclusions | Pre-processing of fH and Tb data |

Although Tb measurements were pre-calibrated to ± 0.2°C during production, the quality of the collected fH measurements depends on the individual-specific signal-to-noise ratio and varies significantly between the loggers. Since the quality index (QI), a measure of the signal-to-noise ratio provided by the logger algorithm is based on all previously taken measurements in each logger, it is not comparable between loggers and therefore requires individual filtering. We used the raw ECG data saved every 60 hours to include only reliable measurements with known uncertainty. We manually calculated the correct bpm for these measurements via the raw ECG trace plots and compared this with the one internally calculated by the logger algorithm. We then individually estimated the assigned error rate for each logger and QI's. We filtered all data accordingly to include only the QI with a known error rate. Furthermore, a manual calculation of all ECGs allowed us to determine the maximum and minimum plausible heart rates that can be observed and verified in the field. After final filtering, we excluded 12 loggers due to insufficient data quality. We expected only measurements with a QI error rate of less than 15% and exhibiting values within the known range of reasonable heart rates. The final data set for analysis included 597,321 and 510,654 measurements of fH and Tb, respectively.

Classification of migration

We used the known breeding site departure and arrival dates for migratory birds recorded via ARU radio telemetry 11 to train a gradient-boosted machine-learning model based on heart rate, body temperature, individual logger ID, individually scaled temperature and heart rate, the difference to the mean heart rate and body temperature, and proportional temperature increase. The model classified all nightly measurements between departure and arrival as migration or stationary phases. Afterwards, we visually classified all measurements by ourselves and compared our manual classification with the one via the machine learning model. Both classifications matched by 0.974% (Model Building AUC 0.966, Classification AUC 0.977). We then used these data to predict arrival on and departure from the wintering sites as well as stopover periods, which were not observable via ARU radio telemetry.

| Reproducibility | To test for differences in heart rate and body temperature for resident and migratory blackbirds in different calendar periods, we used a mixed linear model (R-package 'lme4' 54) with individual measurements of fH / Tb on a resolution of 30 minutes as a response variable and wintering strategy, calendar season, day phase and sex as predictors. |

We performed migration stage-centred analysis with generalised additive models (R-package 'mgvc'), including fH / Tb measurements again as the response variable. Each migration stage was analysed in a separate model, and the days before and after arrival and departure events have been used as a smoothing factor. Wintering strategy and sex were included as predictors. In both analyses, we eliminated temporal autocorrelation, following the established procedure 15,55 of randomly discarding 30% of the data from each individual. In addition, the bird's ID and date were included as random factors to account for individual-specific variation and repeated measurements. We used a post hoc test with a Bonferroni correction to calculate pairwise comparisons in each season.

| Randomization | To analyse energetic differences at different migration stages (see 'Season definitions and individual migration schedules'), we assigned all fH and Tb measurements of resident birds equally to simultaneous measurements of migratory individuals of the same sex. In this way, we avoid pseudoreplication and can compare the physiological data of residents directly to those of migratory blackbirds in relation to their departure and arrival events. This allocation was done multiple times with further analysis to ensure that the observed patterns and differences are consistent. |

| Blinding | As no data was measured manually, no blinding was necessary. All classifications of migration events, as seen in radio telemetry signals and physiological measurement patterns, are always done by two people simultaneously and backed up by statistical models. |

Did the study involve field work?  ☒ Yes  ☐ No

# Field work, collection and transport

| Field conditions | As catching of the birds was done in the field we experienced vaoius weather conditions troughout the study period of three years. After release of the birds at their catching site we monitored environmental conditions like reported in prevouis published work (Linek et al. 2021, DOI: 10.1098/rstb.2020.0213) |

| Location | Mixed forest in southern Germany (47.7801ºN, 9.0203ºE) , Altitude: 690.1 meter above sea level |

| Access & import/export | The field site is a public forest, and we obtained permission from the landowner to capture birds in this area. |
|---|---|
| Disturbance | Animals fitted with radio transmitters and implanted with loggers were allowed to roam freely in their natural habitat, thereby being exposed to all typical and unforeseen environmental disturbances. |

# Reporting for specific materials, systems and methods

We require information from authors about some types of materials, experimental systems and methods used in many studies. Here, indicate whether each material, system or method listed is relevant to your study. If you are not sure if a list item applies to your research, read the appropriate section before selecting a response.

## Materials & experimental systems

| n/a | Involved in the study |
|---|---|
| ☒ | ☐ Antibodies |
| ☒ | ☐ Eukaryotic cell lines |
| ☒ | ☐ Palaeontology and archaeology |
| ☐ | ☒ Animals and other organisms |
| ☒ | ☐ Clinical data |
| ☒ | ☐ Dual use research of concern |
| ☒ | ☐ Plants |

## Methods

| n/a | Involved in the study |
|---|---|
| ☒ | ☐ ChIP-seq |
| ☒ | ☐ Flow cytometry |
| ☒ | ☐ MRI-based neuroimaging |

## Animals and other research organisms

Policy information about studies involving animals; ARRIVE guidelines recommended for reporting animal research, and Sex and Gender in Research

| Laboratory animals | Our study did not involve laboratory animals |
|---|---|
| Wild animals | Study area and captures
We captured a total of 118 adult common blackbirds (Turdus merula) in a mixed forest in southern Germany (47.7801ºN, 9.0203ºE) over three consecutive years (2016-2018). This population is partially migratory - about 26% of all individuals migrate in autumn (female 36%, male 16%) 11. Adult blackbirds (both sexes) were caught with mist nets, fitted with an aluminium leg ring, and transported in cloth cages (height: 30 cm, width: 26.5 cm, length: 49.5 cm) to the Max Planck Institute of Animal Behaviour, Radolfzell (~10-minute drive). At the institute, we surgically implanted internal heart rate and body temperature loggers (see below) and affixed external radio transmitters. Birds were then returned to their original capture location and released.

Surgery
We placed birds on a 40 ° C heating pad to prevent hypothermia and then anaesthetised them with isoflurane (CPH Pharma CP 1 ml/ml, %5). We continuously controlled the bird's body temperature and monitored its breathing frequency. We injected 2 ml of ringer solution into the femur tibia joint to avoid extensive dehydration. After carefully removing some abdominal feathers, we made a 10mm abdominal incision in the skin and tissue layer beneath the sternum. Star-Oddi DST micro-HRT/temperature data loggers (Star-Oddi Ltd., Gardabear, Iceland, dimensions; 8.3 mm x 25.4 mm, weight: 3.3 gram), which were gas sterilised with ethylene-oxide at 38°C before (done at Osypka AG, Rheinfelden, Germany), were inserted after which skin and tissue were stitched separately with an absorbent suture. We then monitored the recovery of the bird in hand and, after ensuring its well-being and normal behavior, we attached a backpack with a radio transmitter (≤ 2.6 g; produced by 1. Sparrow Systems, Fisher, IL, USA, 2. the Swiss Ornithological Institute, Sempach, Switzerland, or 3. Holohil Systems Ltd., Canada) via a leg-loop harness to the bird. The mean weight of a blackbird is about 90 gr, thus radio tags add approximately 5.44% (4.9 gr) to total body mass. To provide some recovery time after surgery and to transport the bird back to the catching site, we placed birds back in a cloth cage where water and food were available ad libitum. In 2015, we conducted a pilot study with five blackbirds kept in aviaries to test their response to implanted loggers and to verify the physical health and wound healing of birds after this type of surgery. Furthermore, during the main study, survival and migratory return rates were not different for birds with implanted loggers compared to rings recovered or radiotagged birds from the previous seven years.
Recapture
We attempted to recapture all birds for data extraction during the following spring. We used the telemetry-derived positions of the birds (either on-site throughout the winter or whose return was recorded by the ARUs) to precisely target recapture using mist nets. After surgical extraction of the data loggers (using the same protocols as for implantation), the birds were released at their capture site. |
| Reporting on sex | Only adults but both sexes were included. The age and sex of captured birds were determined based on differences in the colouration of plumage and beak. |
| Field-collected samples | The study did not involve samples brought from the field into the laboratory. Data was directly measured in the field. |
| Ethics oversight | This work was approved by the responsible ethic commission and ministry in Germany: Regierungspräsidium Freiburg, 35-9185.81/G-16/115, 35-9185.81/G-13/29, 35-9185.81/G-09/08 |

Note that full information on the approval of the study protocol must also be provided in the manuscript.

## Plants

Seed stocks

*Report on the source of all seed stocks or other plant material used. If applicable, state the seed stock centre and catalogue number. If plant specimens were collected from the field, describe the collection location, date and sampling procedures.*

Novel plant genotypes

*Describe the methods by which all novel plant genotypes were produced. This includes those generated by transgenic approaches, gene editing, chemical/radiation-based mutagenesis and hybridization. For transgenic lines, describe the transformation method, the number of independent lines analyzed and the generation upon which experiments were performed. For gene-edited lines, describe the editor used, the endogenous sequence targeted for editing, the targeting guide RNA sequence (if applicable) and how the editor was applied.*

Authentication

*Describe any authentication procedures for each seed stock used or novel genotype generated. Describe any experiments used to assess the effect of a mutation and, where applicable, how potential secondary effects (e.g. second site T-DNA insertions, mosiacism, off-target gene editing) were examined.*

