## [Peer Review File · Nature Ecology & Evolution]

Peer Review Information

Journal: Nature Ecology & Evolution

Manuscript Title: Migratory lifestyle carries no added overall energy cost in a partial migratory songbird

Corresponding author name(s): Nils Linek

Editorial Notes:

Reviewer Comments & Decisions:1
**Decision Letter, initial version:**

7th March 2024

*Please ensure you delete the link to your author homepage in this e-mail if you wish to forward it
to your co-authors.

Dear Dr Linek,

Your manuscript entitled "Migratory lifestyle carries no energy cost in songbirds" has now been
seen by 3 reviewers, whose comments are attached. The reviewers have raised a number of
concerns which will need to be addressed before we can offer publication in Nature Ecology &
Evolution. We will therefore need to see your responses to the criticisms raised and to some
editorial concerns, along with a revised manuscript, before we can reach a final decision regarding
publication.

We are committed to providing a fair and constructive peer-review process. Do not hesitate to
contact us if there are specific requests from the reviewers that you believe are technically
impossible or unlikely to yield a meaningful outcome.

When revising your manuscript:

* Include a "Response to reviewers" document detailing, point-by-point, how you addressed each
reviewer comment. If no action was taken to address a point, you must provide a compelling
argument. This response will be sent back to the reviewers along with the revised manuscript.

* If you have not done so already please begin to revise your manuscript so that it conforms to our
Article format instructions at <http://www.nature.com/natecolevol/info/final-submission>. Refer also
to any guidelines provided in this letter.

* Include a revised version of any required reporting checklist. It will be available to referees (and,
potentially, statisticians) to aid in their evaluation if the manuscript goes back for peer review. A
revised checklist is essential for re-review of the paper.

Please use the link below to submit your revised manuscript and related files:

**[REDACTED]**

**Note:** This URL links to your confidential home page and associated information about manuscripts
you may have submitted, or that you are reviewing for us. If you wish to forward this email to co-
authors, please delete the link to your homepage.

We hope to receive your revised manuscript within four to eight weeks. If you cannot send it within
this time, please let us know. We will be happy to consider your revision so long as nothing similar
has been accepted for publication at Nature Ecology & Evolution or published elsewhere.

Nature Ecology & Evolution is committed to improving transparency in authorship. As part of our
efforts in this direction, we are now requesting that all authors identified as 'corresponding author'
on published papers create and link their Open Researcher and Contributor Identifier (ORCID) with
their account on the Manuscript Tracking System (MTS), prior to acceptance. ORCID helps the
scientific community achieve unambiguous attribution of all scholarly contributions. You can create

352 and link your ORCID from the home page of the MTS by clicking on 'Modify my Springer Nature
account'. For more information please visit please visit www.springernature.com/orcid.

Please do not hesitate to contact me if you have any questions or would like to discuss these
revisions further.

We look forward to seeing the revised manuscript and thank you for the opportunity to review your
work.

**[REDACTED]**

Reviewers' comments:

Reviewer #2 (Remarks to the Author):

General

This study represents a landmark in the study of migratory birds. Collecting many months of heart
rate and body temperature data in migrant and resident individual songbirds is a first, and there
are some very important insights in the paper. That said, we feel that the most exciting parts of
the dataset are physiological, and that more care needs to be made when extending the results of
heart rate and body temperature to energy expenditure (modelled) and to fitness and the evolution
of migratory strategies and tactics. There are clear physiological differences between the migrants
and non-migrants, but how closely do those equate to energy and fitness?

One important potential factor that is not considered is whether seasonal, migration-related
changes in heart size could decouple the heart rate from O₂ consumption and energy expenditure.
Other studies show that heart size can increase during migration. This can affect ventricle volume
and thus stroke volume, so a bird could possibly pump more blood per heart beat and thus a lower
heart rate could equate to the same energy expenditure. It's unfortunate that we do not know
what is going on with heart size in this population, but it could confound the results throughout the
entire year. One message of the paper could be that more detailed physiological studies on the
resident and migrant winter ranges should be done.

Fig 1b is a bit misleading because at first glance it makes the reader think these are the locations
of birds in the study. So the figure legend could be clearer. Moreover, if you don't know where
exactly they went, how do you know the ambient temperature at the wintering area? There are
many things happening here in the estimation of ambient temperature on the migrant's wintering
area that could affect the results. Why did the authors choose to use a single wintering range for
weather rather than estimating the distance each individual travelled from flight duration and a
bearing to the centroid of the known wintering area? Why not put a line southwest from the
breeding site, use flight duration to estimate winter location for each bird, and then get the
weather data from there?

We think that the migration HRT and Tb results are actually more interesting than the estimated
winter energy expenditures from modelling, which could be flawed either due to the model being
unrealistic or other physiological changes in body composition happening. Here you have excellent
evidence of the anticipatory physiological adjustments that individuals make many weeks in
advance of migratory departure. This shows that "decision" to be a migrant this year and the gene
expression, hormonal, metabolic, and behavioural adjustments are made well in advance by some
individuals in a population without a cue like photoperiod. There is a paragraph in the discussion
sort of addressing this but the overall message of the paper is about fitness differences and more
ultimate causation based on fitness which is not actually what was measured.

4105
The authors don't really discuss the fact that residents dropped Tb more in winter than migrants or
the fact that there is a seasonal decrease at all. This Tb and HRT data in itself is very interesting
and relates to how these resident birds are coping with surviving winter, but it is not discussed.

The blackbird partial migrant system is great and many insights may be gained. However, we feel
that the authors may be overemphasizing the results from a short distance migrant. Would these
be enhanced or absent in a long-distance migrants? There could be very different mechanisms.

The title needs to be changed to better reflect the actual study. To say that "Migratory lifestyle
carries no energy cost in songbirds" is inaccurate and misleading. First, all activities have an
energy cost. Second, this title is generalizing to all songbirds from a short distance partially
migratory species of fairly large body size. It is far from the final word on the topic. Maybe what
would be more appropriate would be that "Migration energy costs are compensated by reduced
thermoregulatory costs in a partial migrant songbird population". Of course, emphasizing
"individual-level anticipatory physiological changes weeks before departure in a partially migrant
songbird" would also be a contender.

Perhaps the ultimate causation of partial migration should be answered by measuring survival and
reproduction and making a life table. Trying to do it with physiology alone may not make sense,
although the physiology adds a very interesting dimension to the story.

Below we review in the order introduction, methods, results, discussion.

Specific:

1) Line 12: I can't help but visualize many exceptions from this general statement which really only
applies to bird migration from high to low latitudes. So you either need to specify that or be more
generic, like "Seasonal bird migration may provide energetic benefits associated with moving to
areas with less physiologically challenging climates or increased food availability, but migration
itself can carry high costs".

2) L 15-16: I think you need to also include that you relied on physiological modelling here:
"...paired with automated radio telemetry and physiological modelling..." Or you could say energetic
modelling.

3) L17: Delete "surprisingly" and replace "birds" with "blackbirds". I think your results are very
interesting, but whether they can already be generalized to all birds may be premature without
further study.

4) L19: I think here you need to specify that the model you have predicts energy expenditure
would be decreased enough to compensate for migration costs. So, maybe change to "...departure,
which modelling suggests will dwarf the energy costs...".

5)L 19-20: Change to "...migrants did not appear to decrease total daily energy expenditure..."

6) L26-27: I suggest changing evolved to evolves since it is believed to be an ongoing process.

7) L31: delete temporary and replace the colon with "through".

8) L34: Is this what theory predicts? Doesn't theory generally focus on maximizing fitness or some
reasonable correlate of fitness (Lifetime Reproductive Success or the product of survival and
reproduction)? I don't think LRS is necessarily the same as energy expenditure. An animal could
move to a less energetically costly place, but have lower fitness due to other factors like amount of

food, competition for food, or predation risk. Specifically regarding partial migration, isn't it thought
to be either a mixed ESS where migrants and non-migrants get the same fitness, or a best of a bad
job where residents achieve greater fitness than migrants and migrants switch to residents if
possible? What is the case in blackbirds? Does your data suggest at least for energy the costs are
about equal between tactics?

9) L 38: We suggest rephrasing from "conclusive theory" to something like "greater understanding
of the energetic trade-offs of migration".

10) L40: Singular/plural don't agree. Change to "The Common blackbird is a wide-ranging
species..."

11) L45: Perhaps be more specific about what you mean by conditions getting too harsh. Do you
mean declining temperatures, decreasing food availability, high snowfall?

12) L 46: I think the figure legend of Fig 1b needs to be more explicit that these wintering locations
are not those of the birds in the current study, but those of birds from an earlier geolocator study.
When a reader first looks at the figure they may easily assume that these are the known wintering
locations of study birds.

13) L 51, 56, 135, 189, 200, 617 and elsewhere: Change energetic to energy. Energetics is a field
of study, but grammatically "energetic" also is a modifier. So I think it is better to say what is
being expended: energy.

14) L57: Perhaps change to "...would on average exhibit..."

15) L 445, 449, 452: gram, g, gr – be consistent in notation.

16) L 452: The total weight of the logger (3.3g) and radio tag (2.6g) does not add up 4.9g. It
should be 6.2g or 6.9% of a 90g bird. Perhaps you can make a point that part of the possible
energy/survival costs of external tags is the aerodynamic effects they can have rather than just
extra mass. So, the 3.3 g implanted tag likely has less of a negative cost to the bird's activities,
and you have compared return rates of blackbirds with loggers and tags to those with tags only
(L458). Also, for the future, there are lighter (1g) radio tags available that could have met your
needs for tag life.

17) L 454: ad libitum should be italics.

18) L 491: change to loggers.

19) L 515: If there was only a data quality issue with the heart rate data, why did you end up with
more heart rate measurements than Tb when they were taken at the same rate (every 30 min)?

20) L 526: Is this 97.4% or only 0.974 %?

21) L 527-528: If you could estimate flight durations and stopovers, why not create an estimate of
total migration distance along the known southwest migration route of your previously studied
birds? See me general comment about this. If effect maybe you could have estimated a wintering
area and done a better job at estimating the temperature conditions individuals experienced rather
than using a single estimated temperature regime.

22) L 604: I think you could be clearer here that temperatures were measured in the resident
wintering area, but predicted for the migrants by assuming they were somewhere in the range of
known wintering areas.

 23) L 612: Change to "...from geolocator-based estimates of winter range from previously-studied
 birds..."

 24) L 73: What is meant by higher movement expenses here? Do you mean energy expended on
 movement as inferred from HRT data? Because you follow this by saying that heart rate did not
 differ. Please be more specific.

 25) L 82-83: Does the higher T_b confer other benefits such as better immune function?

26) L 90-94: Another interpretation is that the model predictions are wrong and may be based on
 too many assumptions. Physiological changes in body composition (organ and shivering muscle
 sizes) or mitochondria could also happen that explain why a simple model predicts a difference in
 energy expenditure, while heart rate data do not. Again larger or smaller hearts could pump
 different amounts of blood per beat. In any case I think there should be a case made for more
 detailed physiological studies of the two types.

 27) L 96-126: To me this is the most exciting finding of the study. It's a missed opportunity that
 you start be emphasizing how organ sizes could change, but then do not consider previous
 literature that shows that heart size and thus stroke volume could increase during this
 premigration phase. This would decouple heart rate from oxygen consumption (energy
 expenditure) to some degree. Never the less the data speaks for itself. It clearly suggests that
 metabolic rate (T_b and HRT) at night is decreased in individuals that are preparing to migrate.

 28) L 171-173: Could the nocturnal temperatures at the wintering area have been warmer during
 the spring premigration preparation phase than they were during the fall premigration phase?
 Perhaps it was just warmer and so it wasn't possible for birds to drop T_b and HRT?

 29) L 187: Please be specific that this was estimated from the thermoregulation model, which
 could be flawed. How about "According to the metabolic simulation model we used, on average..."

 30) L 196-199: Why not estimate the energy costs of the migratory flights from allometric
 equations or the "FLIGHT" program. If you know how long they fly then you should be able to
 calculate an approximate number of kJ expended on migratory flight to compare with the
 thermoregulation savings. For example, to fly 800 km at 10m/sec would take 80,000 sec. Using
 the equation in Bishop and Butler 2015 $y=52.6M^{0.74}$ (M in kg) a 90 thrush expends 8.9 W (J/s),
 so the flights would require 712 kJ. That's only 4% of the thermoregulatory difference.

 31) L 230: Missing second parenthesis)

 32) L 235: Change to "energy allocation"

 33) Fig 2 – letters should align with relative position of each subfigure: a above c,d,e.

Reviewer #3 (Remarks to the Author):

 This manuscript reports a study that incorporated implanted heart rate and temperature data
 logger and automated radio telemetry to understand the physiology of migratory birds, making use
 of a partially migratory species – the European blackbirds (*Turdus merula*). By means of these
 technologies, the authors were able to examine the metabolic rates (using heart rate as a proxy)
 and thermoregulation (using body temperature as an indicator) of migratory and non-migratory

7264 individuals, from after breeding to wintering, including both stages of autumnal and vernal
migrations. The obtained data showed internal differences in heart rates and body temperature
between migrants and residents, being most pronounced in certain life-history stages of a year.
More importantly, despite the 5-8 °C higher temperature experienced by the migrants in the
wintering grounds and therefore lower energetic demands for thermoregulation, the physiological
data do not show uni-directional differences between migrants and residents. The complex
differences post a challenge for the authors to interpret the data, yet probably reveal part of the
complex system animal ecologists are routinely dealing with.

I personally admire this study. The methods the authors used (the combination of implanted data
logger and automated radio telemetry) and the data they obtained are absolutely amazing. The
fact that the authors were able to identify migratory versus non-migratory individuals within a
partially migratory population is already rarely achieved to my knowledge, let alone obtaining
longitudinal physiological data at fine temporal scales. The data obtained by this study are
therefore extremely valuable and must be made known to the scientific community, especially for
animal ecologists who study migration ecology and physiology.

That being said, I do have some suggestions for the authors to consider, by which I sincerely hope
that my comments would be considered constructive. Mostly, my comments are questions from a
fellow scientist who is also studying migration physiology in birds. I sincerely hope that my
comments represent the questions that readers may also have and might help the authors to see
some blind spots that may exist in the manuscript.

Major comments:

The major comments I have are about the interpretation and also the presentation of the data,
which are related to each other. For some data, I am wondering whether there might be alternative
explanation. As for the data presentation, although I assumed that the authors had hierarchical
data at both between- and within-individual levels at hands, the figures are unclear to me whether
the trends were observed at which level. Let me elaborate my concerns/questions below:

1. Lines 77-83: In this paragraph, the authors stated that, despite higher ambient temperature
(T_a) in the wintering grounds (Fig. 1c), migrants did not realise “an overall metabolic benefit”
(Lines 77-78). This was the authors’ interpretation to the lack of clear difference in heart rates
between migrants and residents throughout the year (particularly in winter). The authors explained
that this might be due in part to the higher body temperature (T_b) in the migrants (Lines 78-79,
Fig. 2b and 2j) and this “challenges the assumption that migrants realise a thermoregulatory
benefit of overwintering in warmer areas or if they simply expend equivalent energy to maintain
slightly higher T_b .”

I do not outright disagree with the authors’ interpretation on the data. However, I was wondering,
could this result actually reveal the thermoregulatory benefits the migrants have enjoyed in a
warmer wintering area?

We know that heat loss increases when the difference of T_b and T_a increases. Therefore, in a
colder region, maintaining a slightly higher T_b would require more energy and increases heat loss.
Therefore, maintaining a slightly higher T_b might only be affordable in a relatively warmer region.
The question is whether maintaining a slightly higher T_b confers any physiological or survival
benefits to the migrants so that they would spend the energy they could otherwise spare in the
warmer wintering grounds to maintain a slightly higher T_b .

2. Fig. 3: These are amazing data, without a doubt, but I do not fully understand what data the
authors were actually presenting. It would be great if the authors could clarify. This will also help
readers to understand your findings.

First of all, since you have to recapture the birds to retrieve the implanted data logger, I suppose you have very fine within-individual recordings. Did you use them to calculate and obtain a single data point per individual? If so, I may have missed such a description.

Second, if you did not calculate a single data point per individual and instead included the repeated recordings per individual in your analysis (which might be so since you mentioned using hierarchical GAMs), this figure apparently did not represent that kind of data.

I am not criticizing this. My point is, what do the data points on this figure represent? It is now unclear to me whether each of the data points represents one individual at the departure time from the breeding site.

At last, the above questions matter in terms of the interpretation of your results. Although I may have misunderstood the figure, I am highly intrigued whether the reduction of heart rate and body temperature in migrants was observed across individuals or within individuals?

If each data point represents one individual, this pattern could be driven by a correlation between heart rate and departure time, e.g., if late goers have lower heart rates and body temperature than early goers, but this correlation wasn't clear in residents (since they do not go). Could this be a possibility?

3. Overall, I have some concerns about how the authors reached the main conclusion as claimed in the title "Migratory lifestyle carries no energy cost in songbirds." I am not saying that I consider the data insufficient to support this conclusion. My concern is, how this conclusion was reached was unclear to me. After reading this manuscript, I saw the amazing data that showed differences in heart rates and Tb between migrants and residents, and also the lack of differences between the two groups of individuals. However, do these results suggest that migratory lifestyle carries no energy cost in the migrants? Perhaps, but the reasoning and deduction was unclear to me.

Beside my major comments, below are a few minor comments for the authors to consider:

1. Line 90 "Fig. 4": If you reported and cited this figure earlier, why wasn't it "Fig. 3"?
2. Line 150 "Fig. 3c": should this be "3d" since it is about Tb?
3. Lines 174-175: Here I have a minor question about the interpretation. If you attributed the Tb increase of the residents to seasonal changes, why did the migrants lack such a seasonal change?
4. Line 206: would a reference for the Extended POL be needed here?
5. Line 384 in the legend of Fig. 1 "an implanted and temperature logger": should it be "an implanted temperature logger" or "an implanted heart rate and temperature logger"?
6. Lines 451-452: Yet, the internal data logger also adds another 3.3g. Does that not need to be taken into account because it is internal? I am sure that it is still within the weight limit a blackbird can carry without apparent negative impact, but it would be great to clarify whether the weight of an internal data logger needs to be taken into account., e.g., for animal welfare consideration.
7. Lines 457-458: I was exactly curious about the survival and return or recapture rates of the birds that were installed with a data logger. Was that reported in the Results? I did not see it. I would suggest reporting these results, at least in the Extended Data, if not yet.
8. Lines 495-496: That was an amazingly high recapture rate that made this system and the data rare and highly valuable!
9. Lines 584-588: I do not fully understand how you avoided pseudoreplication this way. Could you elaborate this?

Reviewer #4 (Remarks to the Author):

9370

This is a novel and important study that provides new insights into the energetics of short-range avian migration. However, several of the major conclusions depend heavily on estimates of energy expenditure from a biophysical model about which very little detail is provided. At the very least, the authors need to provide full details (most likely in supplementary materials) of the values used to parameterize the NicheMapR model, the source for each value and assumptions, including those about where the birds were roosting, whether digestion-associated thermogenesis substitutes for thermoregulatory heat production, etc.

I am also concerned that the species-specific NicheMapR model that provides the basis for all the estimates of metabolic rate upon which the major conclusions of the study are based was not validated against empirical, species-specific metabolic rate and body temperature data (along the lines of Conradie et al. 2023 JEB doi:10.1242/jeb.245066, in the context of hotter conditions). The NicheMapR endotherm model is still new, and has not, to the best of my knowledge, been properly validated in the context of shallow reductions in rest-phase body temperature. The energy savings associated with small reductions in body temperature can be substantially larger than expected based on the magnitude of hypothermia, and so I am wary about accepting the authors' estimates of thermoregulatory energy expenditure at face value.

Just to be clear, heart rate data were not converted to metabolic rates? It might be worth adding a sentence or two explaining why not, since this has often been done in previous studies.

Clarification is needed on the combined mass of the Star Oddi loggers and backpack-mounted transmitters. In line 445, the mass of each logger is given as 3.3 g, and in line 449 the mass of the backpack transmitters as ≤ 2.6 g. This is a combined mass of 5.9 g, equivalent to 6.6 % of the blackbirds' 90-g body mass, not 4.9 g and 5.4 % as currently stated in line 452. The generally accepted upper limit for attached devices on birds is 5 % of body mass, and I think the authors need to carefully consider whether their data and variables like migration speed or duration may have been influenced?

The title overstates the significance of the study to the point of being misleading. The study involved one species of European partial migrant in which some individuals migrate 800 km, but the title makes it sound like you have shown a lack of an energetic cost for a migratory lifestyle among migratory passerines in general, some of which fly many thousands of kilometers.

Abstract: please remove adjectives like "solid" and "critical"... this is not good scientific writing style.

Lines 40-45: Is there any information available on how the proportion of blackbirds migrating has changed in recent decades with factors like urbanization and climate warming? If there is, please add a sentence or two here.

*****END*****

Author Rebuttal to Initial comments

Referee 1:

This study represents a landmark in the study of migratory birds. Collecting many
420 months of heart rate and body temperature data in migrant and resident individual
songbirds is a first, and there are some very important insights in the paper. That
said, we feel that the most exciting parts of the dataset are physiological, and that
more care needs to be made when extending the results of heart rate and body
temperature to energy expenditure (modelled) and to fitness and the evolution of
migratory strategies and tactics. There are clear physiological differences between
the migrants and non-migrants, but how closely do those equate to energy and
fitness?

***Thank you for commenting on our manuscript and referencing the novelties and***
***uniqueness of our study. We agree that the most interesting parts are the newly***
***gathered data. With the help of your valuable feedback on other potential***
***causes for the physiological differences found between migratory strategies, we***
***concentrated more on those aspects of our findings. We made the model more***
***transparent to clarify its uncertainty. The revised version is considerably more***
***careful in drawing conclusions about energy and fitness differences between***
***the two migratory strategies.***

One important potential factor that is not considered is whether seasonal,
migration-related changes in heart size could decouple the heart rate from O₂
consumption and energy expenditure. Other studies show that heart size can
increase during migration. This can affect ventricle volume and thus stroke volume,
so a bird could possibly pump more blood per heart beat and thus a lower heart rate
could equate to the same energy expenditure. It's unfortunate that we do not know
what is going on with heart size in this population, but it could confound the results
throughout the entire year. One message of the paper could be that more detailed
physiological studies on the resident and migrant winter ranges should be done.

***Thank you very much for this comment. In the revised version, we included***
***corresponding literature (lines 120,128) showing the potential effect of***
***physiological changes in heart rate size in relation to a migratory lifestyle.***
***Additionally, we added a paragraph highlighting the uncertainty of our drawn***
***conclusions (line 248-254) and suggest more morpho- and physiological***
***measurements in the field as a future line of research (lines 219-224, 254).***

11

Fig 1b is a bit misleading because at first glance it makes the reader think these are
the locations of birds in the study. So the figure legend could be clearer. Moreover, if
you don't know where exactly they went, how do you know the ambient temperature
at the wintering area? There are many things happening here in the estimation of
ambient temperature on the migrant's wintering area that could affect the results.
Why did the authors choose to use a single wintering range for weather rather than
estimating the distance each individual travelled from flight duration and a bearing
to the centroid of the known wintering area? Why not put a line southwest from the
breeding site, use flight duration to estimate winter location for each bird, and then
get the weather data from there?

***You are absolutely right about the inaccuracy of the text in the legend. We***
***rewrote the figure legend to clarify the data's origin (line 522-532). Regarding the***
***estimation of ambient temperature at the wintering sites of migratory***
***blackbirds, we have chosen the current approach since the migration direction***
***of blackbirds (from several populations) is quite variable, even within***
***populations. To illustrate this, we have attached a figure of blackbird migration***
***tracks we recorded from all over Europe in the last two years (see Fig. RL1***
***below). We feel that this approach better guarantees that we estimate ambient***
***temperature from observed overwintering locations rather than simulating***
***migratory movements themselves, which could result in predicted***
***overwintering locations with no empirical validation.***

***Using flight durations quantified with heart rate data is an interesting approach,***
***but we feel again that it may be unlikely to improve the accuracy of the***
***estimated ambient temperatures at the wintering sites. The migration direction***
***for individual birds is unknown in this study (and could range between 270***
***degrees west and 180 degrees south). Also, factors that could influence***
***direction and rate of travel (e.g., wind support) cannot be suitably estimated.***

***Overall, we still think, that the ambient temperature values we used are the***
***among-individual mean temperatures (where individual temperatures are***
***recorded at known wintering locations from a prior study on the same***
***population) most accurate.***

***The reason why we intended to include such estimates of ambient temperature***
***from winter sites of migratory blackbirds even if not exact in this study is two-***
***fold. First to get an idea how much warmer wintering sites of migrants may be***

**compared to the sites of their resident conspecifics. And second, to estimate**
**strategy-specific thermoregulatory energy expenditures using the endotherm**
**models. Due to the potential inaccuracy of the estimates of ambient**
**temperature, we run also models including ambient temperatures at the**
**wintering and breeding sites 1 and 2 °C warmer and cooler as the used**
**estimates and found our conclusions were robust to bias in estimates of winter**
**ambient temperatures. This has also been included in the supplementary**
**(Extended Data Fig. 4). Additionally, we want to highlight that these estimates of**
**the ambient temperature of winter sites were not used for any analyses of heart**
**rate or body temperature.**

**Fig. RL1 | Migration tracks and positions of common blackbirds in Europe. The map illustrates the**
**migration paths of blackbirds tracked across Europe over the years 2022 and 2023. Each line**
**represents the track of individual birds, showcasing their diverse routes and stopover points.**

We think that the migration HRT and Tb results are actually more interesting than the503 estimated winter energy expenditures from modelling, which could be flawed either
due to the model being unrealistic or other physiological changes in body
composition happening. Here you have excellent evidence of the anticipatory
physiological adjustments that individuals make many weeks in advance of
migratory departure. This shows that “decision” to be a migrant this year and the
gene expression, hormonal, metabolic, and behavioural adjustments are made well
in advance by some individuals in a population without a cue like photoperiod. There
is a paragraph in the discussion sort of addressing this but the overall message of
the paper is about fitness differences and more ultimate causation based on fitness
which is not actually what was measured.

***Thank you for your comment. We fully agree with the reviewer’s suggestion to***
***put a stronger focus on the physiological outcomes of the heart rate and body***
***temperature data. We have reduced emphasis on the model results throughout***
***the manuscript and focused more on the physiological findings, including***
***moving the results earlier in the manuscript (lines 80-98) and introducing the***
***simulation model later (line 207-224). We also stronger highlighted the***
***anticipatory nature of these physiological adjustments (lines 109-143). Both the***
***physiological data and the model results show that migrants realise the***
***thermoregulatory benefits of higher Body temperature. The model’s primary***
***purpose is to define the expected differences in energy expenditure due to***
***thermoregulation and show support for our initial hypothesis (that migrants***
***expend less energy on thermoregulation). We agree that the model relies on***
***many assumptions, and we have changed many parts of the manuscript to***
***clarify and emphasise these assumptions. However, the results found from a***
***previous study¹ and the model simulations go in the same direction as our***
***trustworthy physical measurements, which we now emphasise more in the***
***revised version.***

The authors don’t really discuss the fact that residents dropped Tb more in winter
than migrants or the fact that there is a seasonal decrease at all. This Tb and HRT
data in itself is very interesting and relates to how these resident birds are coping
with surviving winter, but it is not discussed.

***In the revised version, we now point out more clearly that the reduction of body***
***temperature and seasonal pattern of body temperature and heart rate data is***

*the main scope of our earlier work, already published¹ (lines 168, 193).*

The blackbird partial migrant system is great and many insights may be gained.
However, we feel that the authors may be overemphasizing the results from a short
distance migrant. Would these be enhanced or absent in a long-distance migrants?
There could be very different mechanisms.

*Thank you for this question. We, of course, can only speculate about the*
*implications for a long-distance migrant. However, we did a preliminary sub-*
*analysis of the data in this study where we compared the physiological*
*adjustments between migrants with more and migrants with fewer migration*
*nights (a coarse proxy for travel distance). However, no significant differences*
*were found between the groups, although this may, of course, be due to the*
*smaller sample size. Throughout the discussion, we revised the text to be more*
*careful in our interpretation of the results and changed the title accordingly.*

The title needs to be changed to better reflect the actual study. To say that
“Migratory lifestyle carries no energy cost in songbirds” is inaccurate and
misleading. First, all activities have an energy cost. Second, this title is generalizing
to all songbirds from a short distance partially migratory species of fairly large body
size. It is far from the final word on the topic. Maybe what would be more appropriate
would be that “Migration energy costs are compensated by reduced
thermoregulatory costs in a partial migrant songbird population”. Of course,
emphasizing “individual-level anticipatory physiological changes weeks before
departure in a partially migrant songbird” would also be a contender.

*We agree that the previous title might have sounded like we overstated our*
*conclusions. We previously tried to highlight our intention by using the phrase*
*“net” energy costs in the manuscript. However, to further clarify and avoid any*
*confusion for the reader, we changed the title to “Migratory lifestyle carries no*
*overall energy cost in a partial migratory songbird” and want to thank you for the*
*valuable suggestions on this matter. Also see the previous comment.*

Perhaps the ultimate causation of partial migration should be answered by
measuring survival and reproduction and making a life table. Trying to do it with
physiology alone may not make sense, although the physiology adds a very
interesting dimension to the story.

***We agree that physiology can only be a part of the story of why partial migration***
***evolved. Many other factors, like survival² and reproductive fitness, are crucial.***
***The latter (reproduction) is still missing for this species, largely owing to high***
***predation rates. Here, our findings imply an energy benefit to migrants that may***
***have life history implications – we have revised our discussions to clarify the***
***hypotheses generated by our findings as a useful future course of study (lines***
***257-272, 294-299).***

1. Line 12: I can't help but visualize many exceptions from this general
statement which really only applies to bird migration from high to low
latitudes. So you either need to specify that or be more generic, like
"Seasonal bird migration may provide energetic benefits associated with
moving to areas with less physiologically challenging climates or
increased food availability, but migration itself can carry high costs".

***We agree and changed it according to your suggestion (line 13).***

2. L 15-16: I think you need to also include that you relied on physiological modelling
here: "...paired with automated radio telemetry and physiological modelling..." Or
you could say energetic modelling.

***We agree and added "energetic modelling" to be more specific from the***
***start (line 19).***

3. L17: Delete "surprisingly" and replace "birds" with "blackbirds". I think your results
are very interesting, but whether they can already be generalized to all birds may be
premature without further study.

***Thank you for spotting these imprecisions. We changed it according to your***
***suggestions (lines 13-27).***

4. L19: I think here you need to specify that the model you have predicts energy

expenditure would be decreased enough to compensate for migration costs. So,
maybe change to “...departure, which modelling suggests will dwarf the energy
costs...”.

***Thank you for your comment. However, our model does not quantify the amounts of***
***energy saved through premigratory energy conservation. While our results clearly***
***show a decrease in heart rate during this period, indicating energy savings, we did***
***not directly compare these potential savings to the actual flight costs or later***
***energy amounts from thermoregulatory benefits. We have revised the sentence to***
***clarify this point and to emphasise the uncertainty regarding the extent to which***
***these savings balance against the subsequent costs (line 21).***

5. L 19-20: Change to “...migrants did not appear to decrease total daily energy
expenditure...”

***Thank you for your comment. We changed the wording (line 22).***

6. L26-27: I suggest changing evolved to evolves since it is believed to be an
ongoing process.

***Thank you for your suggestion. We agreed and changed it (line 30).***

7. L31: delete temporary and replace the colon with “through”.

***Thank you for your suggestion. We rewrote the sentence based upon your***
***suggestion (line 35).***

8. L34: Is this what theory predicts? Doesn't theory generally focus on maximizing
fitness or some reasonable correlate of fitness (Lifetime Reproductive Success
or the product of survival and reproduction)? I don't think LRS is necessarily the
same as energy expenditure. An animal could move to a less energetically costly
place, but have lower fitness due to other factors like amount of food,
competition for food, or predation risk. Specifically regarding partial migration,

17

isn't it thought to be either a mixed ESS where migrants and non-migrants get
the same fitness, or a best of a bad job where residents achieve greater fitness
than migrants and migrants switch to residents if possible? What is the case in
blackbirds? Does your data suggest at least for energy the costs are about equal
between tactics?

***Thank you for your question and highlighting the crucial distinction between energy***
***expenditure and fitness outcomes. Our study initially focuses on the energy costs***
***associated with migration, recognising that these do not directly equate to fitness.***
***Fitness indeed involves many complex factors, including- but not limited to - energy***
***expenditure. Here, we started from the energetic perspective to disentangle***
***migratory behaviour's various drivers and components. For blackbirds, specifically,***
***our data indicates a balance in the energetic costs between migrants and residents***
***over winter. This could imply an evolutionary stable strategy, like you already***
***suggested, where neither group consistently outperforms the other in fitness. We***
***acknowledge that fitness is the ultimate measure of success. Still, our primary***
***focus is on understanding whether an energy deficit needs to be overcome through***
***modifications in life history. If an energy deficit exists, it would imply differential***
***fitness or differences in life history strategies. We are currently further investigating***
***repeatability data regarding the wintering strategy and looking deeply into telomere***
***samples from individuals before and after winter to determine the "actual costs" of***
***either strategy. However, we are very aware of the fact that reproductive success is***
***still missing, as mentioned earlier. We appreciate your insightful feedback and have***
***revised the manuscript in multiple sections better to reflect the complex***
***relationship between energy expenditure and fitness, ensuring we do not imply***
***direct fitness conclusions from energetic data alone (e.g., lines 246, 259-272, 274-***
***289).***

9. L 38: We suggest rephrasing from "conclusive theory" to something like "greater
understanding of the energetic trade-offs of migration".

***Thanks a lot for the comment. During our revision, the sentence including***
***"conclusive theory" was completely replaced (lines 39-42)***

10. L40: Singular/plural don't agree. Change to "The Common blackbird is a wide-
ranging species....".

**Thank you for your comment. We corrected the sentence based on your**
 **comment (line 44).**

11. L45: Perhaps be more specific about what you mean by conditions getting too
 harsh. Do you mean declining temperatures, decreasing food availability, high
 snowfall?

**Yes, rapidly declining temperatures leading to ground frost and**
 **continuous snowfall covering any feeding spots appear to be the main**
 **factor that leads to sudden winter migration. In a past preliminary**
 **analysis, we also show some evidence that high snowfall in the breeding**
 **region directly correlates to the percentage of blackbirds leaving**
 **spontaneously in the middle of the winter (see Fig. RL2 below). We revised**
 **the sentence to make what we meant (line 48) clearer.**

Fig. RL2 | Relationship between snow coverage and migration of male blackbirds. This scatter plot depicts the impact of winter snow coverage (measured in centimetres of snow) on the proportion of male blackbirds

19684 *that migrate during the same winter. Data points represent different years, illustrating the correlation*
*between increased snow coverage and a higher percentage of migratory male blackbirds.*

12. L 46: I think the figure legend of Fig 1b needs to be more explicit that these
wintering locations are not those of the birds in the current study, but those of
birds from an earlier geolocator study. When a reader first looks at the figure
they may easily assume that these are the known wintering locations of study
birds.

***Thank you for your comment. We are sorry if there has been any confusion***
***about the origin of those wintering locations and glad you pointed it out.***
***We changed the figure legend to make it clear that the positions belong to***
***birds of the same breeding population of previous studies but not to the***
***birds of the current study. We have also revised the method section in this***
***aspect (lines 521-532, 714).***

13. L 51, 56, 135, 189, 200, 617 and elsewhere: Change energetic to energy.
Energetics is a field of study, but grammatically “energetic” also is a modifier.
So I think it is better to say what is being expended: energy.

***Thank you for your comment. We corrected it throughout the manuscript***
***based on your valuable suggestion.***

14. L57: Perhaps change to “...would on average exhibit...”.

***Thank you for your comment. We changed the wording to be more precise***
***(line 66).***

15. L 445, 449, 452: gram, g, gr – be consistent in notation.

***Thank you for pointing out the inconsistency in the notation regarding the***
***measurement units for weight. The notation for grams has been***

*standardised to 'g' throughout the manuscript.*

16. L 452: The total weight of the logger (3.3g) and radio tag (2.6g) does not add up
 4.9g. It should be 6.2g or 6.9% of a 90g bird. Perhaps you can make a point that
 part of the possible energy/survival costs of external tags is the aerodynamic
 effects they can have rather than just extra mass. So, the 3.3 g implanted tag
 likely has less of a negative cost to the bird's activities, and you have compared
 return rates of blackbirds with loggers and tags to those with tags only (L458).
 Also, in the future, lighter (1g) radio tags will be available to meet your needs for
 tag life.

***Thank you for your comment and for identifying the discrepancy in our***
 ***weight calculations. Indeed, the combined weight of the logger (3.3g) and***
 ***radio tag (2.6g) should total 5.9g, which equates to 6.56% of a 90g bird. We***
 ***double-checked and corrected these numbers (line 594) in the revised***
 ***manuscript. We want to point out that the weight of the transmitter varied***
 ***from around 1.8g to 2.6g and we ensured that the heavier birds got the***
 ***heavier tags of any batch, which mitigates the relative burden. In the***
 ***revised manuscript, we acknowledge that besides the mass, the***
 ***aerodynamic effects of external tags could impact bird activities. By their***
 ***location within the body, the implanted loggers likely have reduced***
 ***aerodynamic influence, contributing to lesser negative impacts on the***
 ***birds. This factor has been crucial in comparing return rates between***
 ***blackbirds with only transmitter backpacks and those with additional***
 ***implanted loggers.***

L 454: ad libitum should be italics.

***Thank you for this comment. We changed it (601).***

17. L 491: change to loggers.

***Thank you for this comment. We did so (line 643).***

18. L 515: If there was only a data quality issue with the heart rate data, why did you
end up with more heart rate measurements than Tb when they were taken at the
same rate (every 30 min)?

***Thank you for spotting this discrepancy. The order of the two numbers was wrong.***
***We corrected it in the revised version (line 666).***

19. L 526: Is this 97.4% or only 0.974 %?

***Thank you for spotting this error. Really, the Classification AUC 0.977 reflects a***
***match of 97.7%. We corrected it (line 677).***

20. L 527-528: If you could estimate flight durations and stopovers, why not create
an estimate of total migration distance along the known southwest migration
route of your previously studied birds? See me general comment about this. If
effect maybe you could have estimated a wintering area and done a better job at
estimating the temperature conditions individuals experienced rather than using
a single estimated temperature regime.

***Thank you for your insights. We acknowledge the potential advantages of***
***the method you proposed for estimating total migration distance and***
***consequent wintering area temperatures. However, as detailed in our***
***general response to your comment (above), the extensive variation in***
***migration directions, even within the same population, led us to adopt a***
***different approach. By averaging the ambient temperatures of all known***
***centroids of wintering locations, we aimed to minimise the error that***
***might arise from assuming a single migration route or stopover for each***
***bird. Additionally, our method mitigates the potential inclusion of non-***
***representative areas that are equidistant but not ecologically relevant to***

22779 ***the blackbirds. We believe that this approach provides a reliable average***
***estimate that captures the typical conditions the population encounters.***
***We also recognise the merit of exploring alternative methods and will***
***consider these for future analyses to compare their precision and***
***applicability.***

21. L 604: I think you could be clearer here that temperatures were measured in the
resident wintering area, but predicted for the migrants by assuming they were
somewhere in the range of known wintering areas.

***We rewrote the sentence to clarify the origin of the temperature data (line***
***769).***

22. L 612: Change to “...from geolocator-based estimates of winter range from
previously-studied birds...”

***Thank you for this comment. We added your suggestion in the revised version (line***
***779).***

23. L 73: What is meant by higher movement expenses here? Do you mean energy
expended on movement as inferred from HRT data? Because you follow this by
saying that heart rate did not differ. Please be more specific.

***We revised this section to refer to the actual energy expenses (derived***
***from heart rate) more accurately during migration (Fig. 3c, Extended Data***
***Fig. 3) (line 93). Furthermore, in the revised version, we clearly emphasise***
***that although the heart rates differ temporarily in different migration***
***stages, no differences can be found overall (e.g., lines 81, 95).***

24. L 82-83: Does the higher Tb confer other benefits such as better immune
function?

***Thank you for this question. Increased immune functions and higher predator***
***avoidance capabilities are among the first benefits we thought of. We added this in***
***our manuscript to elaborate on the benefits (lines 215-219).***

25. L 90-94: Another interpretation is that the model predictions are wrong and may
be based on too many assumptions. Physiological changes in body composition
(organ and shivering muscle sizes) or mitochondria could also happen that
explain why a simple model predicts a difference in energy expenditure, while
heart rate data do not. Again larger or smaller hearts could pump different
amounts of blood per beat. In any case I think there should be a case made for
more detailed physiological studies of the two types.

***Thank you for your comment. We fully agree that other changes in body or***
***organ composition not analysed here could influence their total energy***
***consumption. Instead, the model is based solely on body temperature***
***and ambient temperature. We have stressed this throughout the***
***manuscript's revised version (e.g., lines 210, 563). We have also followed***
***your suggestion and highlighted the necessity of more detailed***
***physiological studies (e.g. measuring body compositions of migrants on***
***their wintering sites) (lines 219-224,248-254).***

26. L 96-126: To me this is the most exciting finding of the study. It's a missed
opportunity that you start be emphasizing how organ sizes could change, but
then do not consider previous literature that shows that heart size and thus
stroke volume could increase during this premigration phase. This would
decouple heart rate from oxygen consumption (energy expenditure) to some
degree. Never the less the data speaks for itself. It clearly suggests that
metabolic rate (Tb and HRT) at night is decreased in individuals that are
preparing to migrate.

***Thank you for your comment. We agree that this previously unknown***
***downregulation of heart rate and body temperature before migration is of great***
***importance. To give more weight to this finding and to address physiological***
***changes more in line with your suggestion, we included more existing literature***

*about morphological adaptations in organ size in the revised version (lines 128-130,*
*but also see previous comment)*

27. L 171-173: Could the nocturnal temperatures at the wintering area have been
warmer during the spring premigration preparation phase than they were during
the fall premigration phase? Perhaps it was just warmer and so it wasn't
possible for birds to drop Tb and HRT?

***Thank you for your question. Yes, that could very well be the case. Additionally, it***
***could also very well be that the whole physiological system is already in a different***
***mode in spring than in fall, with the reproductive apparatus starting to be active to***
***be fully ready as soon as the birds arrive back at their breeding site³ and have to***
***compete with resident individuals for the same territories⁴. We added these points***
***in the revised paragraph version (lines 193-196).***

28. L 187: Please be specific that this was estimated from the thermoregulation
model, which could be flawed. How about "According to the metabolic
simulation model we used, on average..."

***We changed the sentence according to your suggestion (line 227).***

29. L 196-199: Why not estimate the energy costs of the migratory flights from
allometric equations or the "FLIGHT" program. If you know how long they fly
then you should be able to calculate an approximate number of kJ expended on
migratory flight to compare with the thermoregulation savings. For example, to
fly 800 km at 10m/sec would take 80,000 sec. Using the equation in Bishop and
Butler 2015 $y=52.6M^{0.74}$ (M in kg) a 90 thrush expends 8.9 W (J/s), so the flights
would require 712 kJ. That's only 4% of the thermoregulatory difference.

***Thank you for this suggestion. While these calculations would be***
***relatively broad considering uncertainties such as actual distances***
***travelled, current location wind conditions, and altitude, we have added***
***this to the revised version (line 242, also see Methods 'Energy expenditure***

25878 *of migratory flights' and Supplementary Data Table 12). This addition*
*strengthens our argument by providing a more comprehensive*
*understanding of the energy trade-offs involved in migratory behaviour.*

30. L 230: Missing second parenthesis).

*Thank you for spotting the missing parenthesis. We corrected it (line 283).*

31. L 235: Change to “energy allocation”.

*Thank you for this comment. We changed it accordingly (line 288).*

32. Fig 2 – letters should align with relative position of each subfigure: a above c,d,e.

*We initially tried to align the labels in relative order of their mentioning in the main*
*text but agree that it is confusing for the readers to have them “search” for the*
*correct figure. We changed the labelling based on their position within the figure*
*(Fig. 2).*

**Referee 2:**

This manuscript reports a study that incorporated implanted heart rate and temperature data
logger and automated radio telemetry to understand the physiology of migratory birds, making
use of a partially migratory species – the European blackbirds (*Turdus merula*). By means of
these technologies, the authors were able to examine the metabolic rates (using heart rate as a
proxy) and thermoregulation (using body temperature as an indicator) of migratory and non-
migratory individuals, from after breeding to wintering, including both stages of autumnal and
vernal migrations. The obtained data showed internal differences in heart rates and body
temperature between migrants and residents, being most pronounced in certain life-history
stages of a year. More importantly, despite the 5-8 °C higher temperature experienced by the
migrants in the wintering grounds and therefore lower energetic demands for thermoregulation,
the physiological data do not show uni-directional differences between migrants and residents.
The complex differences post a challenge for the authors to interpret the data, yet probably
reveal part of the complex system animal ecologists are routinely dealing with.

I personally admire this study. The methods the authors used (the combination of implanted
data logger and automated radio telemetry) and the data they obtained are absolutely amazing.
The fact that the authors were able to identify migratory versus non-migratory individuals within
a partially migratory population is already rarely achieved to my knowledge, let alone obtaining
longitudinal physiological data at fine temporal scales. The data obtained by this study are
therefore extremely valuable and must be made known to the scientific community, especially
for animal ecologists who study migration ecology and physiology.

That being said, I do have some suggestions for the authors to consider, by which I sincerely
hope that my comments would be considered constructive. Mostly, my comments are
questions from a fellow scientist who is also studying migration physiology in birds. I sincerely
hope that my comments represent the questions that readers may also have and might help the
authors to see some blind spots that may exist in the manuscript.

Major comments:

The major comments I have are about the interpretation and also the presentation of the data,
which are related to each other. For some data, I am wondering whether there might be
alternative explanation. As for the data presentation, although I assumed that the authors had
hierarchical data at both between- and within-individual levels at hands, the figures are unclear

27935 to me whether the trends were observed at which level.

***Thank you for your comments on our manuscript and for pointing out the***
***innovations and uniqueness of our study. With the help of your valuable feedback,***
***we have substantially revised several portions of the manuscript. Specifically, we***
***have clarified in the text and figure captions that most of our analyses refer to***
***expected within-individual dynamics. We are confident that the manuscript now***
***provides more clarity about the data shown and offers a more well-rounded***
***discussion of conclusions that describe alternative explanations for the patterns***
***we documented.***

1. Lines 77-83: In this paragraph, the authors stated that, despite higher ambient temperature
(T_a) in the wintering grounds (Fig. 1c), migrants did not realise “an overall metabolic
benefit” (Lines 77-78). This was the authors’ interpretation to the lack of clear difference in
heart rates between migrants and residents throughout the year (particularly in winter). The
authors explained that this might be due in part to the higher body temperature (T_b) in the
migrants (Lines 78-79, Fig. 2b and 2j) and this “challenges the assumption that migrants
realise a thermoregulatory benefit of overwintering in warmer areas or if they simply expend
equivalent energy to maintain slightly higher T_b .”

I do not outright disagree with the authors’ interpretation on the data. However, I was
wondering, could this result actually reveal the thermoregulatory benefits the migrants
have enjoyed in a warmer wintering area?

We know that heat loss increases when the difference of T_b and T_a increases. Therefore, in
a colder region, maintaining a slightly higher T_b would require more energy and increases
heat loss. Therefore, maintaining a slightly higher T_b might only be affordable in a relatively
warmer region. The question is whether maintaining a slightly higher T_b confers any
physiological or survival benefits to the migrants so that they would spend the energy they
could otherwise spare in the warmer wintering grounds to maintain a slightly higher T_b .

***Thank you very much for your comments and input. Yes, we agree that the***
***thermoregulatory benefits that the migrants realise with apparently similar energy***
***spending could be one of the drivers of this partial migratory system.***

***Thermoregulatory benefits, expressed here in a higher body temperature, could***
***have manifold advantages to individuals. Especially in birds, it has been previously***

28

***shown that higher body temperatures can result in increased immune functions and***
***predator avoidance capabilities. We added examples and references to the revised***
***manuscript to highlight such benefits (lines 217-219). We also fully agree that the***
***mechanism behind these thermoregulatory mechanisms could be driven in***
***significant parts by precisely the concept you described (lines 81-95).***

2. Fig. 3: These are amazing data, without a doubt, but I do not fully understand what data the
authors were actually presenting. It would be great if the authors could clarify. This will also
help readers to understand your findings.

First of all, since you have to recapture the birds to retrieve the implanted data logger, I
suppose you have very fine within-individual recordings. Did you use them to calculate and
obtain a single data point per individual? If so, I may have missed such a description.

Second, if you did not calculate a single data point per individual and instead included the
repeated recordings per individual in your analysis (which might be so since you mentioned
using hierarchical GAMs), this figure apparently did not represent that kind of data.

I am not criticizing this. My point is, what do the data points on this figure represent? It is
now unclear to me whether each of the data points represents one individual at the
departure time from the breeding site.

At last, the above questions matter in terms of the interpretation of your results. Although I
may have misunderstood the figure, I am highly intrigued whether the reduction of heart
rate and body temperature in migrants was observed across individuals or within
individuals?

If each data point represents one individual, this pattern could be driven by a correlation
between heart rate and departure time, e.g., if late goers have lower heart rates and body
temperature than early goers, but this correlation wasn't clear in residents (since they do
not go). Could this be a possibility?

***Thank you for your questions. We are very sorry that our previous figure legend and***
***description within the manuscript weren't clear enough. Indeed, for every***
***individual, we have single-point measurements on a 30-minute interval of varying***

*quality over the entire study from fall to the following spring. However, as we*
*collected data over three years and aimed to compare migrants and residents in*
*different phases of their overwintering cycle, we had to find a way to centre each*
*measurement on a specific time within the migration cycle. To analyse energetic*
*differences at various migration stages (Methods section: ‘Definition of seasons and*
*individual key migratory stages’), we assigned each single heart rate and body*
*temperature measurement of resident birds to simultaneous single measurements*
*of migratory individuals of the same sex based on the real-time timestamp. Due to*
*the complexity of comparing migration phases between the two strategies,*
*especially with no migration for residents, this approach allowed us to effectively*
*synchronise and compare the data from both groups. By distributing all*
*measurements of resident birds (N = 54) from the same sex equally among the*
*migratory birds (N = 19), every single measurement from a resident was only*
*referenced once to a specific measurement of a migrant bird. This resulted in every*
*single measurement of a resident being assigned a “stage of migration”*
*corresponding to the reference migrant measurement, allowing us to directly*
*compare the physiological data of residents and migratory blackbirds in relation to*
*the departure and arrival events of the migrants. Since each resident measurement*
*was assigned only once, the dataset contains unique occurrences of each*
*measurement, thereby avoiding any pseudoreplication.*

*Consequently, Fig. 3 displays the above assigned and centred datasets, with each*
*point being the mean of measurements of all migrants (N = 19) and all residents (N =*
*54) as well relative to specific migration stages (e.g. date of departure.*

*The downregulation of heart rate and body temperature was observed across all*
*individuals, even though it is present in each individual. It is generally masked by a*
*high variation of physiological measurements within and across individuals and*
*time.*

*In response to your feedback, we have revised the manuscript’s methods section*
*(lines 741–751) and figure legend (lines 548–557, i.e. Fig.3) to more accurately*
*describe our approach and the data displayed, ensuring that the physiological*
*trends of heart rate and body temperature are clearly communicated.*

3. Overall, I have some concerns about how the authors reached the main conclusion as
claimed in the title “Migratory lifestyle carries no energy cost in songbirds.” I am not saying
that I consider the data insufficient to support this conclusion. My concern is, how this
conclusion was reached was unclear to me. After reading this manuscript, I saw the
amazing data that showed differences in heart rates and Tb between migrants and
residents, and also the lack of differences between the two groups of individuals. However,
do these results suggest that migratory lifestyle carries no energy cost in the migrants?
Perhaps, but the reasoning and deduction was unclear to me.

***Thank you for your comment. Our original title might have been simplistic. This***
***conclusion of course refers to the equivalent overall energy expenditure implied by***
***equivalent heart rates. That said, we recognise that other parts of the manuscript do***
***indeed show a benefit from, for example, reduced thermoregulatory expense.***
***Thus, in the revised manuscript, we changed the title to be more appropriate:***
***“Migratory lifestyle carries no overall energy cost in a partial migratory songbird”.***
***Additionally, we reworked multiple parts to further elaborate on the limitations of***
***our findings (e.g., lines 33, 80-95, 128, 219-224, 248-254)***

4. Line 90 “Fig. 4”: If you reported and cited this figure earlier, why wasn’t it “Fig. 3”? .

***Thank you for your question. Following a comprehensive revision of significant***
***sections of the manuscript, Fig. 4 has now been properly sequenced after Fig. 3 (line***
***213).***

5. Line 150 “Fig. 3c”: should this be “3d” since it is about Tb?

***Yes, we corrected the reference to the Fig. 3d, which is now Fig. 3h (line 167).***

6. Lines 174-175: Here, I have a minor question about the interpretation. If you attributed the
Tb increase of the residents to seasonal changes, why did the migrants lack such a
seasonal change?

***Thank you for your question. Because the migrants overwinter in more southern***
***regions with a relatively more stable climate, we expect their weather conditions to***
***change less drastically during this time of the year and, therefore, exhibit a more***

**“stable” body temperature¹. We added this disclaimer information to the sentence**
**(lines 171-175).**

7. Line 206: would a reference for the Extended POL be needed here?

**Thank you for pointing out the missing reference. We added it in the revised version**
**of the manuscript (line 259).**

8. Line 384 in the legend of Fig. 1 “an implanted and temperature logger”: should it be “an
implanted temperature logger” or “an implanted heart rate and temperature logger”?

**Thank you for spotting the error. The latter was correct, and we added the word**
**“heart rate” (line 523, i.e. Fig. 1).**

9. Lines 451-452: Yet, the internal data logger also adds another 3.3g. Does that not need to
be taken into account because it is internal? I am sure that it is still within the weight limit a
blackbird can carry without apparent negative impact, but it would be great to clarify
whether the weight of an internal data logger needs to be taken into account., e.g., for
animal welfare consideration.

**Thank you for your comment. The previously stated sum of weight needed to be**
**corrected, and the combined weight of the logger (3.3g) and radio tag (2.6g) should**
**total 5.9g, which equates to 6.56% of a 90g bird. The transmitter weight varied from**
**around 1.8g to 2.6g, and during tagging, we ensured that the heavier birds got the**
**heavier tags of any batch. We acknowledge that besides the mass, the aerodynamic**
**effects of external tags could significantly impact bird activities. Because of their**
**implanted nature, the loggers likely have less of an aerodynamic influence,**
**contributing to a lesser negative impact on the bird’s flight performance and**
**mitigating the relative burden. Our comparisons of return rates between blackbirds**
**with external transmitters only and the combination with an implanted logger have**
**also shown no signs of a higher impact of the combination of radio-transmitter /**
**heart rate logger. Although we would not want to make any general statements**
**about the effect of implanted devices, we can, at least, say that we could not find a**
**negative influence on the migratory propensity, survival and return rate for our**

32

***blackbirds with an addition of an implanted logger and an average weight addition of***
***a maximum of 6.59%.***

10. Lines 457-458: I was exactly curious about the survival and return or recapture rates of the
birds that were installed with a data logger. Was that reported in the Results? I did not see
it. I would suggest reporting these results, at least in the Extended Data, if not yet.

***Thank you for your inquiry. We have added these numbers to the revised Methods***
***section (lines 603-609).***

11. Lines 495-496: That was an amazingly high recapture rate that made this system and the
data rare and highly valuable!

***We appreciate you acknowledging our enormous effort in fieldwork during the***
***recapture periods.***

12. Lines 584-588: I do not fully understand how you avoided pseudoreplication this way.
Could you elaborate this?

***Thank you for your question, and please excuse the inaccuracy of our description of***
***this analysis. By assigning each measurement from a resident bird to a unique***
***reference measurement of a migrant bird, we did not duplicate any comparisons***
***during the analysis. Please see also my response to your comment #2. We revised***
***the paragraph of this description in the revised manuscript to better reflect the***
***purpose of analysing the data on energetic differences at different migration stages***
***and, in particular, between the two phenotypes' overwintering strategies where the***
***residents do not show migration behaviour. We also added more detailed***
***information on the data preparation (lines 741-760).***

**Referee 3:**

This is a novel and important study that provides new insights into the energetics of short-range
avian migration. However, several of the major conclusions depend heavily on estimates of
energy expenditure from a biophysical model about which very little detail is provided. At the
very least, the authors need to provide full details (most likely in supplementary materials) of the
values used to parameterize the NicheMapR model, the source for each value and
assumptions, including those about where the birds were roosting, whether digestion-
associated thermogenesis substitutes for thermoregulatory heat production, etc.

I am also concerned that the species-specific NicheMapR model that provides the basis for all
the estimates of metabolic rate upon which the major conclusions of the study are based was
not validated against empirical, species-specific metabolic rate and body temperature data
(along the lines of Conradie et al. 2023 JEB doi:10.1242/jeb.245066, in the context of hotter
conditions). The NicheMapR endotherm model is still new, and has not, to the best of my
knowledge, been properly validated in the context of shallow reductions in rest-phase body
temperature. The energy savings associated with small reductions in body temperature can be
substantially larger than expected based on the magnitude of hypothermia, and so I am wary
about accepting the authors' estimates of thermoregulatory energy expenditure at face value.

***Thank you very much for reading our manuscript and highlighting the need for***
***greater transparency regarding our use of the NicheMapR model in our study. In***
***response to your concerns, we have extensively revised our manuscript by focusing***
***more on our physiological data measured by the logger and providing detailed***
***descriptions of the parameters used in the model. We are still convinced that our***
***model is a valuable contribution to our analysis as it offers an explanation of how***
***migrants potentially realise energetic benefits even though the heart rate of***
***migrants and resident individuals doesn't differ. Additionally, the model serves as***
***an additional analytical dimension to our heart rate measurements and quantifies***
***our estimated energy sums while aligning well with our empirically measured***
***physiological data. The revised manuscript is now more careful about the***
***conclusions drawn from the model calculations, which are meant simply to***
***approximate the differences in thermoregulatory expense. That said, our ground-***
***truthed physiological measurements also agree with the conclusions drawn from***
***the model. Our previous work, also cited in the paragraphs concerned ¹, illustrates***
***the expected heart rate decrease while in a warmer environment. The formulated***

34

*NicheMapR model is consistent with these previously observed relationships and*
 *tries to give them an approximate magnitude and convert them into kilojoules. Even*
 *if the general assumptions are somewhat off, the large magnitude of the differences*
 *suggests that the energetic benefits of the migratory phenotype are likely*
 *qualitatively valid. In other words, our observations of heart rate, body temperature,*
 *and ambient temperature by themselves (without the model) strongly imply an*
 *energy ‘surplus’ for migrants arising from reduced thermoregulatory burden. The*
 *NicheMapR model is consistent with this expectation and is robust to temperature*
 *deviations of several degrees (see Extended Data Fig. 4). In the revised manuscript,*
 *we have amended sections dealing with model assumptions and results*
 *interpretation to reflect these considerations more accurately, ensuring our*
 *conclusions remain conservative and supported by our data, the model’s*
 *theoretical framework^{5,6}, and previous studies^{1,7}. All to say, we have made a more*
 *explicit acknowledgement that physiological, morphological, or behavioural*
 *differences among the strategies could influence the model estimates, but we note*
 *that these differences would need to be quite substantial to account for the*
 *magnitude of energy differential our findings suggest.*

Just to be clear, heart rate data were not converted to metabolic rates? It might be worth adding
 a sentence or two explaining why not, since this has often been done in previous studies.

*Thank you for your question. Initially, we tried to calibrate the HRT rate measures,*
 *performing cliometrics measurements for each recaptured bird in a metabolic*
 *chamber as soon as we got hold of them. However, due to the difficult recapture*
 *procedure, spread out return dates and the nature of seasonal differences in heart*
 *rate and body temperature (see¹ and previous comment), we couldn’t correctly*
 *calibrate the physiological measurements to metabolic rates. Since this would*
 *technically be just a conversion to another currency without adding any extra*
 *information regarding a comparison of migratory phenotypes, it was omitted during*
 *the process. Critically, our heart rate data suggest equivalent overall energy*
 *expenditure thus conversion to units of energy is not needed to compare*
 *components of this budget because the terms would cancel. The hidden expenses*
 *can also be seen just in the physiological data alone. We added such a disclaimer*
 *and explanation in the revised version (lines 59, 81, 95, 227-232, 248-254)*

Clarification is needed on the combined mass of the Star Oddi loggers and backpack-mounted
transmitters. In line 445, the mass of each logger is given as 3.3 g, and in line 449 the mass of
the backpack transmitters as ≤ 2.6 g. This is a combined mass of 5.9 g, equivalent to 6.6 % of the
blackbirds' 90-g body mass, not 4.9 g and 5.4 % as currently stated in line 452. The generally
accepted upper limit for attached devices on birds is 5 % of body mass, and I think the authors
need to carefully consider whether their data and variables like migration speed or duration may
have been influenced?

***Thank you for pointing out this discrepancy in our calculations. You are correct that***
***the combined weight of the logger (3.3g) and radio tag (2.6g) should total 5.9g, which***
***equates to 6.56% of a 90g bird. The weight of the transmitter varied from around 1.8g***
***to 2.6g, and we ensured that the heavier birds got the heavier tags of any batch,***
***which may mitigate the relative burden. We also acknowledge that besides the***
***mass, the aerodynamic effects of external tags may play an important factor in a***
***bird's flight performance. By their location within the body, the implanted loggers***
***likely have reduced aerodynamic influence, potentially contributing to lesser***
***negative impacts on the birds. This factor has been crucial in our comparison of***
***return rates between blackbirds with only transmitters and those with additional***
***internal loggers. Recapture, and migratory return rates were not lower for birds with***
***implanted loggers compared to only radiotagged birds from previous years. Return***
***rates for birds with implanted loggers were 90%, compared to 43% for the control***
***group, and recapture rates were 80% versus 23%, respectively. The experimental***
***setup may have influenced these findings, which required extensive recapturing***
***efforts to retrieve the loggers and continuous monitoring, making direct***
***comparisons challenging. We added this information in our revised manuscript***
***(lines 592-609).***

The title overstates the significance of the study to the point of being misleading. The study
involved one species of European partial migrant in which some individuals migrate 800 km, but
the title makes it sound like you have shown a lack of an energetic cost for a migratory lifestyle
among migratory passerines in general, some of which fly many thousands of kilometers.

***We agree that the previous title might have been overemphasising the implications.***
***We changed the title to "Migratory lifestyle carries no overall energy cost in a partial***

**migratory songbird” to be more specific and to further clarify and avoid any**
**confusion for the reader (line 1).**

1. Abstract: please remove adjectives like “solid” and “critical”... this is not good scientific
writing style.

**Thank you for your comment. In the revised version, we have removed those**
**adjectives for the sake of clarity and to be more specific from the start on (lines 13-**
**27)**

2. Lines 40-45: Is there any information available on how the proportion of blackbirds
migrating has changed in recent decades with factors like urbanisation and climate
warming? If there is, please add a sentence or two here.

**Thank you for your question. Indeed, studies have investigated the influence of**
**climate change and urbanisation on blackbirds^{8,9}. Current data shows that with**
**increasing urbanisation, blackbirds might become more resident, likely as a result**
**of the different microclimates that exist in cities with ambient conditions being less**
**harsh and additional feeding keeping a constant supply of food availability for birds**
**in these areas. We have added this information in the revised paragraph (line 50).**

Response Letter References

1. Linek, N. *et al.* A songbird adjusts its heart rate and body temperature in response to season and fluctuating daily conditions. *Philosophical Transactions of the Royal Society B: Biological Sciences* **376**, 20200213 (2021).
2. Zúñiga, D. *et al.* Migration confers winter survival benefits in a partially migratory songbird. *eLife* **6**, e28123 (2017).
3. Dominoni, D., Quetting, M. & Partecke, J. Artificial light at night advances avian reproductive physiology. *Proceedings of the Royal Society B: Biological Sciences* **280**, 20123017 (2013).

374. Fudickar, A. M., Schmidt, A., Hau, M., Quetting, M. & Partecke, J. Female-biased obligate strategies in a partially migratory population. *Journal of Animal Ecology* **82**, 863–871 (2013).
5. Kearney, M. R., Briscoe, N. J., Mathewson, P. D. & Porter, W. P. NicheMapR – an R package for biophysical modelling: the endotherm model. *Ecography* **44**, 1595–1605 (2021).
6. Porter, W. P. & Kearney, M. Size, shape, and the thermal niche of endotherms. *Proceedings of the National Academy of Sciences* **106**, 19666–19672 (2009).
7. Linek, N. *et al.* A partial migrant relies upon a range-wide cue set but uses population-specific weighting for migratory timing. *Movement Ecology* **9**, 1–14 (2021).
8. Partecke, J. & Gwinner, E. Increased sedentariness in European blackbirds following urbanization: A consequence of local adaptation? *Ecology* **88**, 882–890 (2007).
9. Berthold, P. A comprehensive theory for the evolution, control and adaptability of avian migration. *Ostrich* **70**, 1–11 (1999).

Decision Letter, first revision:

Our ref: NATECOLEVOL-23123033A

23rd July 2024

Dear Dr. Linek,

Thank you for your patience as we've prepared the guidelines for final submission of your Nature Ecology & Evolution manuscript, "Migratory lifestyle carries no overall energy cost in a partial migratory songbird" (NATECOLEVOL-23123033A). Please carefully follow the step-by-step instructions provided in the attached file, and add a response in each row of the table to indicate the changes that you have made. Please also check and comment on any additional marked-up edits we have proposed within the text. Ensuring that each point is addressed will help to ensure that your revised manuscript can be swiftly handed over to our production team.

We would like to start working on your revised paper, with all of the requested files and forms, as soon as possible (preferably within two weeks). Please get in contact with us immediately if you anticipate it taking more than two weeks to submit these revised files.

38When you upload your final materials, please include a point-by-point response to any remaining reviewer comments.

In recognition of the time and expertise our reviewers provide to Nature Ecology & Evolution's editorial process, we would like to formally acknowledge their contribution to the external peer review of your manuscript entitled "Migratory lifestyle carries no overall energy cost in a partial migratory songbird". For those reviewers who give their assent, we will be publishing their names alongside the published article.

Nature Ecology & Evolution offers a Transparent Peer Review option for new original research manuscripts submitted after December 1st, 2019. As part of this initiative, we encourage our authors to support increased transparency into the peer review process by agreeing to have the reviewer comments, author rebuttal letters, and editorial decision letters published as a Supplementary item. When you submit your final files please clearly state in your cover letter whether or not you would like to participate in this initiative. Please note that failure to state your preference will result in delays in accepting your manuscript for publication.

Cover suggestions

We welcome submissions of artwork for consideration for our cover. For more information, please see our guide for cover artwork.

Nature Ecology & Evolution has now transitioned to a unified Rights Collection system which will allow our Author Services team to quickly and easily collect the rights and permissions required to publish your work. Approximately 10 days after your paper is formally accepted, you will receive an email in providing you with a link to complete the grant of rights. If your paper is eligible for Open Access, our Author Services team will also be in touch regarding any additional information that may be required to arrange payment for your article.

Please note that *Nature Ecology & Evolution* is a Transformative Journal (TJ). Authors may publish their research with us through the traditional subscription access route or make their paper immediately open access through payment of an article-processing charge (APC). Authors will not be required to make a final decision about access to their article until it has been accepted. Find out more about Transformative Journals

Authors may need to take specific actions to achieve compliance with funder and institutional open access mandates. If your research is supported by a funder that requires immediate open access (e.g. according to Plan S principles) then you should select the gold OA route, and we will direct you to the compliant route where possible. For authors selecting the subscription publication route, the journal's standard licensing terms will need to be accepted, including [a href="https://www.nature.com/nature-portfolio/editorial-policies/self-archiving-and-license-to-publish](https://www.nature.com/nature-portfolio/editorial-policies/self-archiving-and-license-to-publish). Those licensing terms will supersede any other terms that the author or any third party may assert apply to any version of the manuscript.

Please use the following link for uploading these materials:
[REDACTED]

[REDACTED]

Reviewer #2:

Remarks to the Author:

Thank you for your thoughtful responses to our questions and for making the appropriate changes to the manuscript. Our only suggestion is that the title still suggests there is no cost of migrating and and so perhaps changing it to Migratory lifestyle carries no ADDED overall energy cost in a partial migratory songbird is appropriate

Reviewer #3:

Remarks to the Author:

In my opinion, the authors did an excellent job in revising this manuscript and the concerns I raised last time were well addressed. I only have some minor editorial suggestions are listed below for the authors to consider:

1. Title: Echoing other reviewers' comments, I was indeed thinking if the title matches with the actual findings of this study. Although the authors have changed the title, I still think that the current title is still simplistic and probably does not accurately convey the gist of the study. However, because the exciting and important findings of this study are pretty nuanced, I find it very difficult to make a title that is succinct and catchy at the same time. So, while I think it would be better to change the title again, I have no idea what to suggest to the authors. My apologies.

2. Lines 25-26: I would suggest slightly rephrasing the sentence to “Moreover ... insights suggesting that the maintenance of migration is associated with...”

3. Lines 38-39 “whereas others may require different life history strategies to overcome energy deficits”: semantically unclear, probably better to rephrase.

4. Lines 54-56: It's not clear without reading the figure legend, perhaps slightly rephrasing it would help. For example, “... experience on average ~5.7 °C warmer ambient temperatures (Ta) over 39 years ...”

5. Tense: Through out the paper, there are many places where I think the authors should have used past tense. I have to admit that tense is not my strong suit, but I nevertheless list those places for the authors to check:

- a. Line 59: examined
- b. Line 60: investigated
- c. Line 61: quantified
- d. Line 124: occurred
- e. Line 129: which we “did” not
- f. Line 133: all birds “were”
- g. Line 152: were
- h. Line 153: migrated
- i. Line 154: travelled
- j. Line 159: migrants' nocturnal Tb “was”
- k. Line 191: this difference “was”
- l. Line 194: did not allow

6. Line 61: I suggest adding “(i.e., migrants versus residents)” after “among phenotypes”

417. Line 63 “negatively colinear with Ta”: not sure if this is a correct way to say it. Would “negatively correlated with Ta” or “covaries with Ta negatively” work here?
8. Line 65: add “thus”, i.e., “We thus predicted...”
9. Line 68 “incur”: how about “bear”?
10. Line 68 “from”: due to?
11. Lines 85-86: This sentence has a long clause “the fact that...” without a follower verb. Perhaps change it to “Moreover, migratory blackbirds exhibited a slightly...”?
12. Line 97 “does”: This word is superfluous. Delete it.
13. Line 98 “various overwintering stages”: What does it mean?
14. Line 147: clearer if saying “an even greater 36.4% increase”
15. Line 168: clearer if adding “that” after “we observed”
16. Line 175: delete “the initial”, not necessary here
17. Line 178 “can already be seen”: could be seen already
18. Line 185 “lead-up to spring migration”: not sure what it meant, perhaps delete “to”?
19. Line 230” delete “is”, i.e., “as implied by”
20. Lines 594-595 “the heavier birds received the heavier tags”: I think the two articles “the” are not necessary here.
21. Line 595 “mass”: weight?
22. Lines 600-602: You conducted a pilot study in 2015 and then what?
23. Lines 604-606: Thank you for providing the information. This is super intriguing as both the return rates and recapture rates were so much higher in tagged birds than in control birds. Perhaps it was due to the huge effort you paid, but would it be a sign of inadvertent biases?

24.Line 746: I suggest slightly rephrasing it as “This assigned each single measurement of a resident a “stage of migration” ...”

Reviewer #4:

Remarks to the Author:

Thanks for your thoughtful responses to the concerns I raised. This is a very exciting study and a substantial contribution to our understanding of avian energetics. Congratulations!

Author Rebuttal, first revision:

Reviewer #2:

Thank you for your thoughtful responses to our questions and for making the appropriate changes to the manuscript. Our only suggestion is that the title still suggests there is no cost of migrating so perhaps changing it to Migratory lifestyle carries no ADDED overall energy cost in a partial migratory songbird is appropriate

Thank you for all your effort and help in improving our manuscript. We agree and adapted your suggested change of the title.

Reviewer #3:

In my opinion, the authors did an excellent job in revising this manuscript and the concerns I raised last time were well addressed. I only have some minor editorial suggestions are listed below for the authors to consider:

Thank you for your immense input, and we are happy that all your concerns could have been solved. Thank you also for those last suggestions to further refine our manuscript's clarity and correct grammatical issues.

1. Title: Echoing other reviewers' comments, I was indeed thinking if the title matches with the actual findings of this study. Although the authors have changed the title, I still think that the current title is still simplistic and probably does not accurately convey the gist of the study. However, because the exciting and important findings of this study are pretty nuanced, I find it very difficult to make a title that is succinct and catchy at the

same time. So, while I think it would be better to change the title again, I have no idea what to suggest to the authors. My apologies.

We agree that this is a difficult one but think that now with the suggestion of reviewer 2 we found a good solution. The revised title is now "Migratory lifestyle carries no added overall energy cost in a partial migratory songbird".

2. Lines 25-26: I would suggest slightly rephrasing the sentence to "Moreover ... insights suggesting that the maintenance of migration is associated with..."

Thank you very much for this comment. We changed it according to your suggestions (lines 25-27).

3. Lines 38-39 "whereas others may require different life history strategies to overcome energy deficits": semantically unclear, probably better to rephrase.

Thank you very much for this comment. We changed it according to your suggestions (lines 37-39).

4. Lines 54-56: It's not clear without reading the figure legend, perhaps slightly rephrasing it would help. For example, "... experience on average ~5.7 °C warmer ambient temperatures (Ta) over 39 years ..."

Thank you very much for this comment. We changed it according to your suggestions (line 54).

5. Tense: Through out the paper, there are many places where I think the authors should have used past tense. I have to admit that tense is not my strong suit, but I nevertheless list those places for the authors to check:

- a. Line 59: examined
- b. Line 60: investigated
- c. Line 61: quantified
- d. Line 124: occurred
- e. Line 129: which we "did" not
- f. Line 133: all birds "were"
- g. Line 152: were
- h. Line 153: migrated
- i. Line 154: travelled

- j. Line 159: migrants' nocturnal Tb “was”
- k. Line 191: this difference “was”
- l. Line 194: did not allow

Thank you for spotting these imprecisions in the use of tense. We changed it according to your suggestions (lines 59, 60, 62, 126, 131, 139, 155, 156, 161, 193, 196).

- 6. Line 61: I suggest adding “(i.e., migrants versus residents)” after “among phenotypes”

Thank you for your suggestion. We agreed and changed it (line 61).

- 7. Line 63 “negatively colinear with Ta”: not sure if this is a correct way to say it. Would “negatively correlated with Ta” or “covaries with Ta negatively” work here?

Thank you for your question. Yes, your suggested phrase works as well. We have changed it based on your suggestion, even though the original phrase was also correct (lines 63 and 64).

- 8. Line 65: add “thus”, i.e., “We thus predicted...”

Thank you for this comment. We changed it (lines 65 and 66).

- 9. Line 68 “incur”: how about “bear”?

We have adopted the proposed change (line 68).

- 10. Line 68 “from”: due to?

Thank you for this comment. We changed it (line 69).

- 11. Lines 85-86: This sentence has a long clause “the fact that...” without a follower verb. Perhaps change it to “Moreover, migratory blackbirds exhibited a slightly...”?

We completely agree that it was a hard-to-read sentence and revised it according to your suggestion (line 86-87).

12.Line 97 “does”: This word is superfluous. Delete it.

You are right. We deleted it (line 98).

13.Line 98 “various overwintering stages”: What does it mean?

Thank you for this comment. We rewrote the sentence and added explanatory examples (lines 99-100).

14.Line 147: clearer if saying “an even greater 36.4% increase”

We added the proposed specification of the direction of change (line 149).

15.Line 168: clearer if adding “that” after “we observed”

Thank you for this comment. We agreed and added the word “that” (line 171).

16.Line 175: delete “the initial”, not necessary here

That is right. We deleted the phrase (line 174).

17.Line 178 “can already be seen”: could be seen already

We changed it to ‘could be seen already’ as your proposed (lines 180-181).

18.Line 185 “lead-up to spring migration”: not sure what it meant, perhaps delete “to”?

Thank you for this comment. We hope it became clearer now with the revised version we deleted the word ‘to’ (line 187).

19.Line 230” delete “is”, i.e., “as implied by”

We changed the sentence according to your suggestion (line 232).

20.Lines 594-595 “the heavier birds received the heavier tags”: I think the two articles “the” are not necessary here.

Thank you for this comment. We deleted the first article but kept the second one because we think it enhances the flow of reading (lines 329-330).

21.Line 595 “mass”: weight?

The word ‘weight’ could have been used too. However, we keep ‘mass’ as we want to be certain here, and weight can be different with active flight acceleration and similar behavioural changes.

22.Lines 600-602: You conducted a pilot study in 2015 and then what?

Thank you for your question. We rewrote the paragraph to clearly state the outcomes of the pilot study. And separated the sentence in two (lines 335-337).

23.Lines 604-606: Thank you for providing the information. This is super intriguing as both the return rates and recapture rates were so much higher in tagged birds than in control birds. Perhaps it was due to the huge effort you paid, but would it be a sign of inadvertent biases?

Thank you for this important comment! However, it wasn’t higher in the tagged ones but in the implanted ones. You are correct about the mentioned bias, and one could argue that this is likely inherent in every type of fieldwork with wild animals. The main purpose of the comparison, however, is to serve ethical considerations and highlight that recapture and survival rates are for sure not lower for our experimental birds with implants.

24.Line 746: I suggest slightly rephrasing it as “This assigned each single measurement of a resident a “stage of migration”...”

Thank you for your suggestion. We agree and changed the sentence according to your proposed way (line 481).

Reviewer #4:

Thanks for your thoughtful responses to the concerns I raised. This is a very exciting study and a substantial contribution to our understanding of avian energetics. Congratulations!

Thank you very much for your previous valuable input and acknowledgement of our work.

Final Decision Letter:

Dear Dr Linek,

We are pleased to inform you that your Article entitled "Migratory lifestyle carries no added overall energy cost in a partial migratory songbird", has now been accepted for publication in Nature Ecology & Evolution.

Over the next few weeks, your paper will be copyedited to ensure that it conforms to Nature Ecology and Evolution style. Once your paper is typeset, you will receive an email with a link to choose the appropriate publishing options for your paper and our Author Services team will be in touch regarding any additional information that may be required

Due to the importance of these deadlines, we ask you please us know now whether you will be difficult to contact over the next month. If this is the case, we ask you provide us with the contact information (email, phone and fax) of someone who will be able to check the proofs on your behalf, and who will be

2available to address any last-minute problems . Once your paper has been scheduled for online publication, the Nature press office will be in touch to confirm the details.

Acceptance of your manuscript is conditional on all authors' agreement with our publication policies (see www.nature.com/authors/policies/index.html). In particular your manuscript must not be published elsewhere and there must be no announcement of the work to any media outlet until the publication date (the day on which it is uploaded onto our web site).

Please note that *Nature Ecology & Evolution* is a Transformative Journal (TJ). Authors may publish their research with us through the traditional subscription access route or make their paper immediately open access through payment of an article-processing charge (APC). Authors will not be required to make a final decision about access to their article until it has been accepted. Find out more about Transformative Journals

Authors may need to take specific actions to achieve compliance with funder and institutional open access mandates. If your research is supported by a funder that requires immediate open access (e.g. according to Plan S principles) then you should select the gold OA route, and we will direct you to the compliant route where possible. For authors selecting the subscription publication route, the journal's standard licensing terms will need to be accepted, including <https://www.nature.com/nature-portfolio/editorial-policies/self-archiving-and-license-to-publish>. Those licensing terms will supersede any other terms that the author or any third party may assert apply to any version of the manuscript.

We welcome the submission of potential cover material (including a short caption of around 40 words) related to your manuscript; suggestions should be sent to Nature Ecology & Evolution as electronic files (the image should be 300 dpi at 210 x 297 mm in either TIFF or JPEG format). Please note that such

pictures should be selected more for their aesthetic appeal than for their scientific content, and that colour images work better than black and white or grayscale images. Please do not try to design a cover with the Nature Ecology & Evolution logo etc., and please do not submit composites of images related to your work. I am sure you will understand that we cannot make any promise as to whether any of your suggestions might be selected for the cover of the journal.

You can generate the link yourself when you receive your article DOI by entering it here: <http://authors.springernature.com/share>.

[REDACTED]

P.S. Click on the following link if you would like to recommend Nature Ecology & Evolution to your librarian <http://www.nature.com/subscriptions/recommend.html#forms>

** Visit the Springer Nature Editorial and Publishing website at www.springernature.com/editorial-and-publishing-jobs for more information about our career opportunities. If you have any questions please click here. **